# Patterns of item nonresponse behaviour to survey questionnaires are systematic and associated with genetic loci

Gianmarco Mignogna ®[1,2,3,4,19], Caitlin E. Carey[1,4,5,19], Robbee Wedow ®[4,6,7,8,9,19] ✉, Nikolas Baya[1,4], Mattia Cordioli ®[2], Nicola Pirastu ®[10,11], Rino Bellocco[3,12], Kathryn Fiuza Malerbi[13], Michel G. Nivard ®[14,15,16], Benjamin M. Neale ®[1,4,5,17,18], Raymond K. Walters ®[1,4,20] & Andrea Ganna ®[1,2,17,20] ✉

Response to survey questionnaires is vital for social and behavioural research, and most analyses assume full and accurate response by participants. However, nonresponse is common and impedes proper interpretation and generalizability of results. We examined item nonresponse behaviour across 109 questionnaire items in the UK Biobank ($N = 360,628$). Phenotypic factor scores for two participant-selected nonresponse answers, 'Prefer not to answer' (PNA) and 'I don't know' (IDK), each predicted participant nonresponse in follow-up surveys (incremental pseudo-$R^2 = 0.056$), even when controlling for education and self-reported health (incremental pseudo-$R^2 = 0.046$). After performing genome-wide association studies of our factors, PNA and IDK were highly genetically correlated with one another ($r_g = 0.73$ (s.e. = 0.03)) and with education ($r_{g,PNA} = -0.51$ (s.e. = 0.03); $r_{g,IDK} = -0.38$ (s.e. = 0.02)), health ($r_{g,PNA} = 0.51$ (s.e. = 0.03); $r_{g,IDK} = 0.49$ (s.e. = 0.02)) and income ($r_{g,PNA} = -0.57$ (s.e. = 0.04); $r_{g,IDK} = -0.46$ (s.e. = 0.02)), with additional unique genetic associations observed for both PNA and IDK ($P < 5 \times 10^{-8}$). We discuss how these associations may bias studies of traits correlated with item nonresponse and demonstrate how this bias may substantially affect genome-wide association studies. While the UK Biobank data are deidentified, we further protected participant privacy by avoiding exploring non-response behaviour to single questions, assuring that no information can be used to associate results with any particular respondents.

Ethics, participant privacy and interpretation of behavioural genetics research are primary concerns for this research team and this study takes particular precautions in these areas. We urge readers to carefully read Box 1, our Ethical Approval statement in Methods and our Frequently Asked Questions document in the Supplementary Information.

Item nonresponse occurs when no substantive answer is recorded for a study participant on a given questionnaire item, such as when the participant does not provide an answer or responds 'I do not know'[1]. Nonresponse is interesting both as a behavioural choice by a survey participant and as a statistical concern due to missing data. Much social and behavioural research relies on surveys, and data analysis of survey data usually assumes full and accurate response by survey participants, or at least that any nonresponse is independent of the outcomes that a researcher is interested in. In reality, nonresponse is common and

## BOX 1

# Ethical considerations

We acknowledge that a dark history of pseudoscientific discrimination and genetic essentialism underlies the study of behavioural outcomes in genetics. Along with this history comes a responsibility for genetics researchers to carefully consider and develop robust ethical protocols when conducting their research, especially in the area of behavioural genetics and sociogenomics. However, we also recognize the extraordinary benefits of using genetics as a powerfully emergent tool to better understand behaviour and its relation to health.

This paper is one such attempt to uniquely evaluate a source of bias that is present in nearly all questionnaire-based research: item nonresponse. We believe the analyses and conclusions of our study not only underscore the importance of how item nonresponse relates to different domains of health and health behaviour, but also push researchers to be highly sceptical that selection biases can be tackled without considering the specific genetic and behavioural sources of selection of the specific data and system under study. In short, because so much research into the genetics of human health and wellbeing is based on surveys, it is important to understand how nonresponse could be impacting the generalizability of genetics results.

Participant consent is critical for the ethical conduct of research. Nonresponse, including at the item level, in some instances will reflect a participant exercising their (entirely justified) right to voluntarily not participate in some aspect of a study. This is especially true in the case of item nonresponse in the form of actively responding 'Prefer not to answer'. As a result, it requires careful ethical consideration to evaluate how to study nonresponse without breaching the participant's consent as reflected in both the item-level nonresponse and the study-level informed consent and participation. There can be ethical harms to ignoring the source of missing data in research. Consideration of missingness is necessary to identify the ways in which a study or a particular analysis may not be representative of the population, otherwise researchers risk the myriad impacts of uncritically producing biased or ungeneralizable results. Reifying incorrect results risks group harm from incorrect conclusions that may reinforce misperceptions of disadvantaged groups. Decades of social science research on mechanisms

of missing data and their influence on research results reflect recognition of the imperative to wrestle with this challenge.

In line with the new guidance on ethical standards announced by Nature Human Behaviour (https://www.nature.com/articles/s41562-022-01443-2; https://www.nature.com/articles/s41562-022-01472-x), we have carefully planned and implemented our ethical approach by: (1) seeking specific approval from the datasets we use to conduct this study; (2) following closely the requirements and evaluations of our Institutional Review Boards (IRBs); and (3) creating and documenting our approach in our Box 1, in our Frequently Asked Questions (FAQ) document and in our methods. Most importantly, because a participant's right to voluntarily not take part in a study is a critical component of ethical research, we avoid exploring nonresponse behaviour to single questions and thus assure that no information can be used to associate results with any particular respondent. We make one exception, in an analysis of responding 'I don't know' to the question 'During your childhood, how many times did you suffer painful sunburn?', where we empirically check that our analysis is meeting the stated goal of avoiding revealing item-specific factors. We intentionally rely on the IDK response (that is, item nonresponse that does not imply a desire to avoid participation in the item) and use an item that is less socially sensitive to minimize ethical concerns while doing this check.

The factors generated for our analyses can be thought of as reflecting a general behavioural tendency for someone to choose not to respond to one or more survey items; they are not reflective of nonresponse to any single, specific item. Methods for correcting non-response bias (including Heckman correction) may implicitly estimate associations of missingness with expected values of the target phenotype, but these interim statistical quantities are neither designed to draw conclusions about any individual nor are they the scientific goal. For this reason, we do not report effect sizes for the stage 1 model in Heckman correction to similarly avoid inference from missingness of individual items (in this case, fluid intelligence (FI)) outside of necessary calculation of intermediate quantities.

We stress that these ethical considerations apply not only within this study, but also in future applications and extensions of this work.

---

the independence assumption often unjustified, impeding proper interpretation and generalizability of results.

Understanding the causes of nonresponse and nonparticipation has long been a concern for survey-based research[2]. As an observable behaviour, nonresponse represents a complex interplay between survey design for questionnaires and a respondent's cognitive processes, that is, in understanding a question and choosing a response[1,3]. Nonresponse at the item level may be thought of as an intermediate behaviour on the spectrum between providing complete data and complete nonparticipation—that is, unit nonresponse[4]—and nonresponse is predictive of future study dropout[5]. Further, nonresponse is unlikely to be captured by a single construct since individuals may differ in their likelihood to select different nonresponse choices in a questionnaire, for example 'I don't know', 'I'm not sure' or 'I don't want to answer', both overall and when responding to questions in certain categories[6].

One primary motivation to understand this nonresponse behaviour is that it not only reduces the effective sample size of scientific studies but can also introduce bias[7]. As a general framework, data may

be either missing completely at random (MCAR), missing randomly conditional on the remaining observed data (missing at random or MAR) or missing dependent on the unobserved data (missing not at random or MNAR)[6,8]. Thus, if there are unobserved individual phenotypic differences influencing the likelihood of item nonresponse, then the resulting missingness is considered MNAR. Nonrandom missingness is of particular concern because common statistical methods such as full information maximum likelihood estimation or multiple imputation are only sufficient to address MAR data. MNAR requires more direct modelling that includes assumptions about the type of missingness[9–11].

In recent years, these concerns about the consequences of voluntary participation in research cohorts have seen renewed interest for large-scale biorepositories such as the UK Biobank (UKB). Several researchers have raised concerns that biobank cohorts may not be representative of the population[12–15]. For instance epidemiological comparisons with UK Biobank show strong evidence of sampling bias[16–18], and correlation of this selection with geography has the potential to increase effects of population stratification[19,20]. Recent research has

demonstrated that the collider bias induced by nonrandom participation can substantially bias associations with genetic variants, creating spurious associations for variants associated with sampling probabilities and reducing generalizability of results outside the studied population[12–14,16,18,21–23]. These concerns have prompted several suggestions for analytic methods to reduce selection bias in genetic studies of large biobanks[13,14,18,24,25], but no consensus solution has emerged thus far.

To understand these concerns about selection bias in genome-wide association studies (GWAS), it is instructive to diagram the potential links between nonresponse, genetics and target outcomes that could induce biased associations (Fig. 1 and ref. 22). Broadly, since analyses of observed data such as the UKB implicitly condition on nonresponse, associations may be biased if this conditioning induces collider bias on the joint distribution of genetic data, covariates and phenotypes. In particular, consider the case of a phenotype that is positively associated with nonresponse or other selection, either directly (that is, $\theta > 0$) or through some mediator. Single nucleotide polymorphisms (SNPs) that increase this phenotype (that is, $\alpha > 0$) will then also be associated with the nonresponse and the observed correlation of the SNP with the phenotype will probably be reduced due to restriction of range. Conversely, SNPs that increase nonresponse without being mediated by the phenotype (that is, $\alpha = 0, \beta > 0$) will become negatively associated with the phenotype among observed samples since increased nonresponse will favour lower expected values of the nonresponse-associated phenotype. By extension, when a pair of traits have this same pattern of confounding (that is, through $\beta$), GWAS of the observed data could yield spurious genetic correlation even if the genetic associations on the two traits (for example, through $\alpha$) are orthogonal. Similar concerns arise for paths mediated through other variables (for example, through $\delta, \gamma$), depending on their relationship with the phenotype of interest (that is, $\eta$).

This potential for confounding by genetic associations with missingness is not just theoretical given that non-response behaviour is clearly correlated with many heritable traits. Higher rates of item nonresponse are associated with lower educational attainment and poorer health status[26–28]. Increased item nonresponse has also been observed for individuals with more depressive symptoms[29] and lower self-confidence, among other psychological and personality traits[30]. These individual differences in item nonresponse rates can however be sensitive to the content of the questionnaire items[26,31] and the characteristics of the study population[28,32]. Still, similar patterns are often observed for unit nonresponse and study attrition; for example, participants with lower educational levels are more likely to drop out of a study[33]. Similarly, those with heavy alcohol habits, higher levels of mental distress and those who abstain from unhealthy behaviours tend to be underrepresented in studies due to their higher attrition rates[34–36].

Recent work has begun to shed light on the genetics of study participation and nonresponse. A longitudinal study in the UK estimated heritability of 18–32% for continued participation after baseline[37]. Polygenic score analyses have also associated unit nonresponse with educational attainment, schizophrenia, personality and smoking, among other heritable phenotypes[36,37]. In the UKB, substantial genetic associations have been observed for participation in follow-up surveys[38] and for sex differences in participation at baseline[39]. A recent GWAS using first-degree relatives also found significant genetic variants associated with likelihood of participating in the UKB at baseline, with substantial genetic correlations observed with educational attainment and body mass index. There remains hope that identifying these genetic components of nonresponse behaviour may assist with modelling MNAR mechanisms in genotyped samples to reduce selection bias[40].

The current study extends this work on nonresponse in biorepositories to evaluate item response bias in the UKB, providing insight on its additional contribution to nonrandom sampling beyond study-level influences on selection and participation at baseline and unit-level nonresponse at follow-up. We first explore the phenotypic structure

of the nonresponse options provided by the UK Biobank in the initial cohort assessment, and we estimate latent factors for a person's general propensity to respond to questionnaires with 'Prefer not to answer' (PNA) or 'I don't know' (IDK). We then perform GWAS of these two factors, identifying significantly associated loci and modelling genetic correlations with other heritable traits. We validate these genetic findings through out-of-sample polygenic prediction of non-response behaviour, and we demonstrate that by modelling the missingness mechanism using auxiliary variables in Heckman correction we can reduce bias from nonresponse that does not depend directly on the missing value. Throughout this investigation, we avoid any analysis that would violate any participant's stated desire to avoid answering a question (Box 1). We anticipate that these findings will provide insight into genetic variants associated with cognitive processes involved in item nonresponse and also provide a basis for evaluating the impact of non-response bias on GWAS of other traits and disorders.

## Results

### Distribution of item nonresponse across questions

To investigate item non-response behaviour, we considered two distinct answer choices from the UK Biobank touchscreen questionnaire: 'Prefer not to answer' and 'I don't know' (PNA and IDK, respectively). The PNA option was available for all 109 analysed questions with non-response options, of which 83 questions allowed the IDK option, administered to the study population of 360,628 unrelated participants of European genetic ancestry with available genetic data. Participants selected PNA less frequently (8.82% at least once) than IDK (67.02% at least once), possibly reflecting the effects of the IDK option being presented first among the response options or effects of social desirability bias or satisficing to avoid attributing a nonresponse to personal preference[41] (Table 1 and Supplementary Fig. 1). For each question, on average, 0.16% of participants chose PNA and 2.17% chose IDK (Supplementary Tables 1 and 2). Importantly, individuals could only select one non-response answer per question, so a response of IDK necessarily precluded a response of PNA and vice versa.

The demographic trends of individuals with at least one non-response were broadly consistent with previous non-response literature[18] (Table 1). Females answered PNA more often than males (9.4% females vs 8.2% males, $P < 5 \times 10^{-5}$), while males were more likely to answer IDK than females (66.0% females vs 68.3% males, $P < 5 \times 10^{-5}$). Nonresponders had markedly lower educational attainment (18.73% of nonresponders had college or university degrees vs 33.45% for responders for PNA; 29.41% of nonresponders had college or university degrees vs 37.75% for responders for IDK).

The items with the highest rates of nonresponse were consistent with differential use of the two nonresponse options. PNA was the more common response among questions capturing potential illegal behaviour or social stigma (for example, 'How often do you drive faster than the speed limit on the motorway?' or 'Does your partner or a close relative or friend complain about your snoring?') (Supplementary Fig. 2a). By comparison, participants selected IDK more frequently in questions about their distant past such as 'Were you breastfed when you were a baby?' or 'During your childhood, how many times did you suffer painful sunburn?', consistent with a higher difficulty of remembering those items (Supplementary Fig. 2b). We hypothesized that the frequency of PNA or IDK answers might also increase as a function of the order in which the questions were asked because of fatigue experienced by the participant as time spent taking a survey increases, but negative binomial regression found no evidence of a positive trend for IDK or PNA (Supplementary Fig. 2).

### Correlation patterns of item nonresponse and factor analyses

To measure the degree to which item nonresponse behaviour is shared across survey questions, we began by inspecting phenotypic (tetrachoric) correlations (Fig. 2). PNA answers showed an overall higher

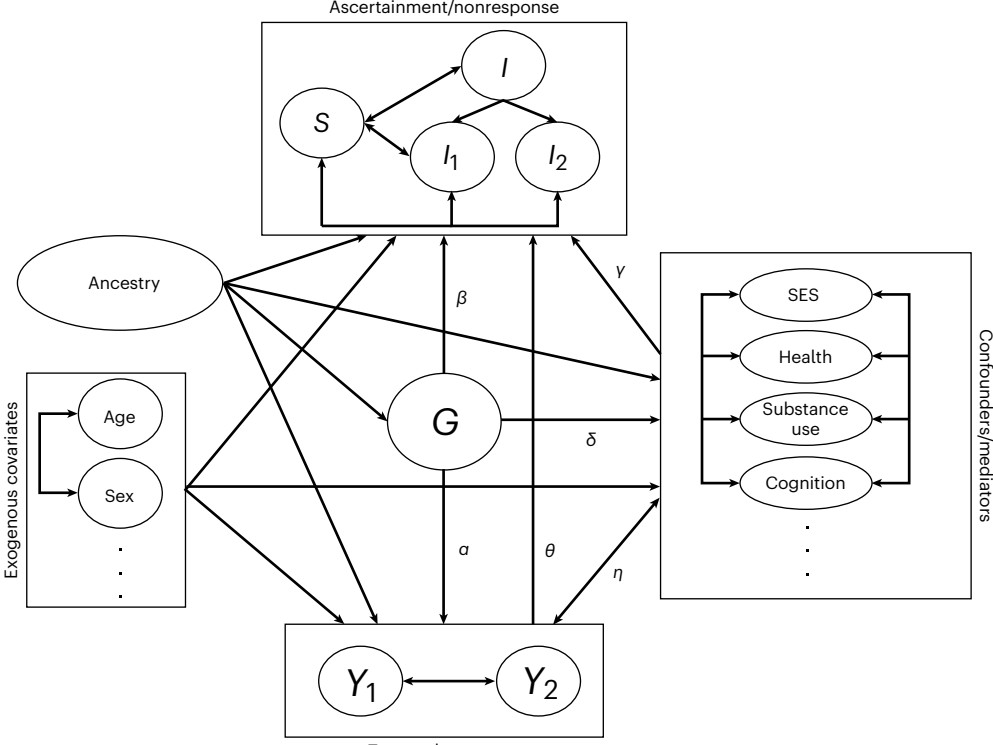

**Fig. 1 | Directed acyclic graph (DAG) of potential influences linking nonresponse to the association between genetics ($G$) and target phenotypes ($Y_1$ and $Y_2$).** Potential elements of nonresponse include overall item non-response behaviour ($I$), item-specific nonresponse ($I_1$ and $I_2$) and survey or study-level nonparticipation or ascertainment ($S$). Boxes indicate sets of traits, with paths to/from a box indicating potential paths to one or more traits in the box. GWAS aims to discover direct ($\alpha$) or indirect ($\delta \times \eta$) associations with genetics $G$ on phenotypes $Y$ conditional on covariates. Analyses of observed data implicitly condition on nonresponse ($I$,$S$) and thus may be biased if that conditioning affects the expected joint distribution of genetic data, covariates and phenotypes (for example, $\beta \neq 0$, $\theta \neq 0$ or $\gamma \times \delta$ or $\gamma \times \eta \neq 0$). Modelling the missingness mechanism, including use of mediators as auxiliary variables, can reduce bias from nonresponse that does not depend directly on the missing value (that is, paths other than $\theta$). The current study demonstrates association of genetics with nonresponse (that is, $\beta$ and $\delta \times \gamma$) and considers the prospect of modelling missingness in GWAS.

correlation than IDK (mean $r^2 = 0.66$ (interquartile range (IQR) = 0.17) for PNA and mean $r^2 = 0.28$ (IQR = 0.13) for IDK), indicating that individuals who responded to questions with PNA tended to do so more consistently across questions than individuals who responded to questions with IDK. Indeed, we identified a small number of individuals who responded to all survey questions with PNA ($N = 11$).

Item non-response behaviour was also more similar among survey questions from similar phenotype domains. In other words, item non-response behaviour was not independent of answering patterns across questions. For example, the average correlation of PNA answers among questions within the food intake and the mental health domains (mean $r^2 = 0.85$ (IQR = 0.08) and mean $r^2 = 0.76$ (IQR = 0.12), respectively) was higher than the correlations between food intake and mental health questions (mean $r^2 = 0.14$ (IQR = 0.05)).

On the basis of the observed structure across survey domains, we next estimated latent (unmeasured) factors for overall IDK and PNA item non-response behaviour, conditional on the correlated sub-structure. To do so, for each response type, we first assessed the survey substructure by performing factor analysis (FA) with the full set of questions and examining cluster analyses of the residual correlations from a single-factor model (Supplementary Figs. 3 and 4, respectively). These residuals provided us with the magnitude of the correlation not explained by a single general factor model, allowing us to identify bifactor FA model as an appropriate model for the survey substructure. The chosen bifactor FA models the observed correlation matrix for item nonresponse as a function of one general factor affecting nonresponse for all items and possibly two or more additional domain-specific

factors affecting subsets of items identified by the model. Since this model may not fully address nested substructure within groups of items, we fit the bifactor FA model on a reduced set of survey questions pruned for high pairwise correlations observed in the residual cluster analysis of the single-factor model (Methods). From exploratory factor analysis, we selected a 5-factor solution for the pruned PNA responses and a 4-factor solution for the pruned IDK responses, both with oblique ('biquartimin') rotations, as our final models based on standard fit metrics (Supplementary Table 3).

The common general latent factor, representing the underlying general item non-response behaviour across questions, explained 51.26% and 25.61% of the total variance for PNA and IDK, respectively, based on the selected models. Our approach also identified sub-stantial variance in item non-response behaviour (11.63% and 11.20% for PNA and IDK, respectively) that was accounted for by additional domain-specific factors rather than a general factor (Fig. 3, for con-firmatory factor analysis (CFA): Supplementary Figs. 5 and 6 for PNA and IDK, respectively). Two of these factors (influencing items we might consider as affecting 'Health' and 'Psychiatric' domains) par-tially overlap between PNA and IDK. The domain-specific factor with items related to 'Ethnicity' was specific to PNA and was present when respondents did not answer questions about ethnic background and skin colour, with loadings of 0.69 and 0.51, respectively.

We focused the remainder of our analysis on the general factors for PNA and IDK since we expect these factors to capture broad predis-position to nonresponse that should be more generalizable than the domain-specific factors and less specific to the set of items included

**Table 1 | Baseline demographics of PNA and IDK nonresponders**

| Characteristic | | No PNA (N=328,843) | PNA (N=31,785) | No IDK (N=118,928) | IDK (N=241,700) |
|---|---|---|---|---|---|
| Mean age (s.d.) - yr | | 56.64 (8.00) | 58.55 (7.84) | 55.49 (8.08) | 57.46 (7.89) |
| Age group - no. (%) | ≤51yr | 93,859 (28.54) | 6,678 (21.01) | 40,217 (33.82) | 60,320 (24.96) |
| | 51<yr≤58 | 77,607 (23.60) | 6,517 (20.50) | 29,220 (24.57) | 54,904 (22.72) |
| | 58<yr≤63 | 80,082 (24.35) | 8,151 (25.64) | 26,638 (22.40) | 61,595 (25.48) |
| | yr>63 | 77,295 (23.51) | 10,439 (32.84) | 22,853 (19.22) | 64,881 (26.84) |
| Female sex - no. (%) | | 175,701 (53.43) | 18,166 (57.15) | 66,000 (55.50) | 127,867 (52.90) |
| Participants in UK Regions - no. (%) | East Midlands | 23,476 (7.14) | 2,419 (7.61) | 8,336 (7.01) | 17,559 (7.26) |
| | London | 21,953 (6.68) | 2,139 (6.73) | 8,340 (7.01) | 15,752 (6.52) |
| | North East | 38,720 (11.77) | 3,990 (12.55) | 13,859 (11.65) | 28,851 (11.94) |
| | North West | 50,212 (15.27) | 5,276 (16.60) | 17,009 (14.30) | 38,479 (15.92) |
| | Scotland | 25,216 (7.67) | 2,416 (7.60) | 9,293 (7.81) | 18,339 (7.59) |
| | South East | 30,905 (9.39) | 2,603 (8.19) | 11,762 (9.89) | 21,746 (9.00) |
| | South West | 45,626 (13.87) | 3,659 (11.51) | 17,246 (14.50) | 32,039 (13.26) |
| | Wales | 13,804 (4.20) | 1,311 (4.12) | 4,981 (4.19) | 10,134 (4.19) |
| | West Midlands | 28,499 (8.67) | 3,186 (10.02) | 10,051 (8.45) | 21,634 (8.95) |
| | Yorkshire | 50,432 (15.34) | 4,786 (15.06) | 18,051 (15.18) | 37,167 (15.38) |
| College/University degree - no. (%) | | 110,011 (33.45) | 5,924 (18.73) | 44,890 (37.75) | 71,075 (29.41) |
| Mean Townsend Deprivation Index (s.d.) | | −1.59 (2.92) | −0.72 (3.34) | −1.72 (2.83) | −1.41 (3.02) |

PNA and IDK columns refer to those participants who chose these options at least once throughout the questionnaire.

in our analysis in UKB, while avoiding ethical concerns from more narrowly targeted inference (Box 1).

### PNA and IDK predict response to follow-up questionnaires

To evaluate the relationship of non-response behaviour with other phenotypic and genetic measures, we estimated each individual's latent factor scores for the general PNA and IDK factors from confirmatory analysis of the selected bifactor models (Supplementary Fig. 7). We hypothesized that the common latent factors for PNA and IDK should relate to future item-level non-response behaviours, as well as maybe capturing a broader non-participation tendency (that is, survey non-response). Therefore, we evaluated whether the PNA and IDK latent factors were able to predict item-level nonresponse on a follow-up mental health questionnaire, as well as whether participants did or did not complete the online follow-up dietary questionnaires distributed by UKB.

Among individuals in our sample who completed the mental health follow-up questionnaire (N = 118,037), 25.3% responded IDK at least once and 10.1% responded PNA at least once across the 58 questions available for analysis. This is a substantially lower rate of ever responding IDK than the baseline (67%), probably reflecting the differential content of the survey and the added opportunities for nonparticipation in the follow-up (for example, ascertainment[33], survey-level nonresponse). PNA and IDK factor scores computed at baseline jointly predicted whether an individual selected a non-response option for at least one of the 58 questions included in the analysis (N = 36,517), beyond basic sample covariates (incremental pseudo-$R^2$ = 0.056). Within this model, each non-response factor contributed significant variance ($\beta_{PNA}$ = 0.336 (s.e. = 0.02), P = 3.60 × 10$^{-83}$; $\beta_{IDK}$ = 0.823 (s.e. = 0.01), P < 1 × 10$^{-308}$). These results were only minimally altered by the inclusion of college completion and self-reported health as additional covariates (incremental pseudo-$R^2$ = 0.046, $\beta_{PNA}$ = 0.316 (s.e. = 0.02), $\beta_{IDK}$ = 0.800 (s.e. = 0.01)), indicating that our estimated non-response factors capture information about item nonresponse that is not fully explained by health and education as previously established correlates of nonresponse[26]. The generalizability of baseline item-level

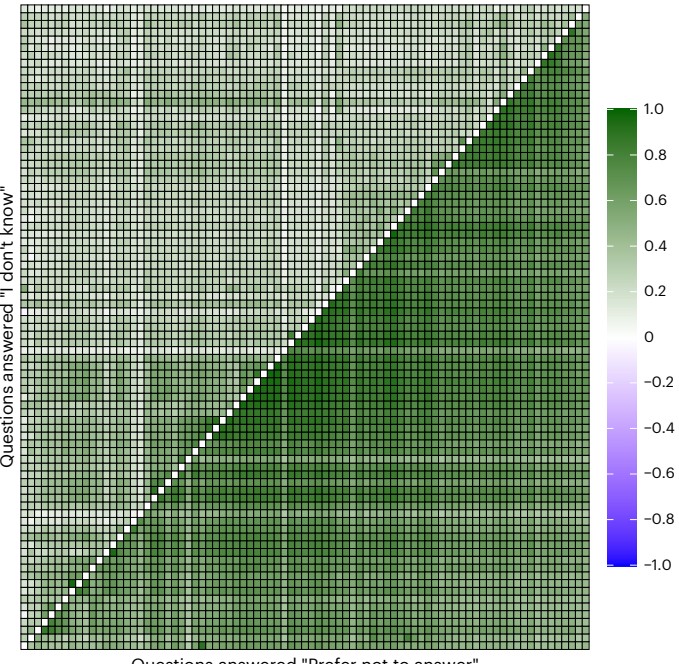

**Fig. 2 | Phenotypic (tetrachoric) correlation of item nonresponse among questions.** Each question has been recoded as dichotomous (1 = PNA or IDK, 0 otherwise). Each row and column represent the same question. We considered only questions that allow both the 'Prefer not to answer' and the 'I don't know' options. Upper triangle, 'I don't know'; lower triangle, 'Prefer not to answer'. The scale of possible correlations ranges from −1 to 1.

non-response behaviour to robustly predict future item nonresponse, albeit with modest accuracy, suggests that item nonresponse is at least partially durable as an individual trait, consistent with potential genetic associations with this behaviour.

To examine prediction of future survey-level response based on our baseline item-level non-response factors, we considered separately the completion of the follow-up online dietary questionnaire only in the first wave of the invitation ($N = 69,735$ completed versus $N = 146,712$ not completed for PNA and IDK analyses, respectively) or in all the 4 waves of the invitation ($N = 19,097$ completed versus N = 99,151 not completed for PNA and IDK analyses, respectively). We quantified the improved prediction after controlling for basic covariates, as well as education and self-rated health status since these have been shown to be proxies for survey non-response behaviours[14,24]. PNA and IDK factors slightly improved prediction of participation for all four waves of survey invitation, on top of education and self-rated health status, both when each factor was considered independently (incremental pseudo-$R^2 = 0.0027$, $P < 2 \times 10^{-16}$ and incremental pseudo-$R^2 = 0.0012$, $P = 3 \times 10^{-11}$ for PNA and IDK, respectively) and when the two factors were combined (incremental pseudo-$R^2 = 0.0034$, $P < 2 \times 10^{-16}$). Combining PNA and IDK resulted in a better prediction of not responding to all four waves of follow-up survey invitations compared with predicting just one wave (incremental pseudo-$R^2 = 0.0014$, $P < 2 \times 10^{-16}$) (Supplementary Table 4). In sum, the general factors for item nonresponse are associated with whether participants will continue to engage in future follow-up research, and our estimated scores for those factors are able to provide prediction beyond established proxies for nonresponse such as education and self-rated health status.

## GWAS of item nonresponse

To assess potential genetic components of item non-response behaviour, we conducted a GWAS on the estimated factor scores for the general PNA and IDK behaviour across survey questions. We identified 4 genome-wide significant ($P < 5 \times 10^{-8}$) loci for PNA and 35 loci for IDK (Fig. 4, and Supplementary Tables 5 and 6, respectively). Of these loci, 2 were shared between PNA and IDK.

Variants within these loci have been previously associated with traits in the domains of health (for example, type 2 diabetes[42]), psychiatry (for example, schizophrenia[43]), personality (for example, neuroticism[44]), cognition (for example, intelligence[45]) and socioeconomic status (for example, educational attainment[46]). The lead SNP of the top locus for IDK, rs9401593 in an intronic region of chromosome 6, has been previously associated with socioeconomic status (SES) traits such as household income[47] and years of education[48], as well as with broader participation-related phenotypes such as providing valid information necessary for recontact and accepting an invitation to complete a mental health follow-up questionnaire[33,38]. Beyond the top locus, several additional IDK loci were also associated with participation phenotypes in previous studies (that is, providing valid recontact information[38], completion of a mental health follow-up questionnaire[33] and sex-differential participation bias[39]). The top locus for PNA, a locus shared with IDK, maps to a highly pleiotropic missense SNP in *SLC39A8* (p.Ala391Thr, rs13107325). Given the wide range of phenotypes associated with this variant—from schizophrenia[49] to Crohn's disease[50] to scoliosis[51]—it is perhaps unsurprising that this locus would be associated with both forms of non-response patterns in this study. Such overlap with GWAS of other traits may reflect some instances of shared aetiology, for example through the many traits associated with health and socioeconomic status, but may also be evidence of genetic variants associated with nonresponse leading to selection bias in some previous GWAS studies.

The general factors for PNA and IDK were both significantly heritable, with higher estimated SNP heritability for IDK ($h^2_g = 0.068$, $P = 3.46 \times 10^{-95}$) than for PNA ($h^2_g = 0.021$, $P = 6.16 \times 10^{-16}$). We also observed a significant heritability enrichment for central nervous system cell types (enrichment $P_{PNA} = 0.001$, $P_{IDK} = 7.32 \times 10^{-17}$)[52], consistent with the expectation that nonresponse relies on cognitive processes[3]. Within the brain[52,53], both PNA and IDK were significantly associated with regions differentially expressed in the cerebellum

(coefficient $P = 0.003$ for both PNA and IDK; Supplementary Table 7). Importantly, the SNP heritability for these factor scores also shows substantially stronger genetic signal than GWAS using a simple sum of the number of nonresponses over all survey items. In comparison, the SNP heritability of the simple sum score for PNA responses was lower and non-significant ($h^2_g = 0.002$, $P = 0.5$). This is consistent with an expectation that the factor analysis provides improved power, reducing measurement error across items and clarifying the signal in the context of correlated residual structure.

## Shared vs question-specific item non-response behaviour

One concern in our analysis is that the GWAS results for the item non-response phenotypes may be driven by questions with the highest number of PNA and IDK responses (Supplementary Fig. 2), rather than capturing an underlying behaviour shared across survey questions. This is a concern both because it affects the interpretation of the results and because it could expose undesired information about nonresponse to individual items (Box 1). To explore this concern, we performed a GWAS of IDK for the question with the largest number of IDK responses, which was 'During your childhood, how many times did you suffer painful sunburn?' We observed a moderate genetic correlation with the IDK factor ($r_g = 0.40(0.03)$, $P = 2.13 \times 10^{-34}$) and we identified 4 genome-wide significant loci for this GWAS. None of these 4 loci were genome-wide significant in the GWAS of either the PNA or IDK factors. Instead, the top genetic results appear to correlate with the number of sunburn occasions. For example, the IDK-increasing allele for rs35407 allele, a 3' UTR variant in SLC45A2, is associated with increased risk for melanoma[54] and cutaneous squamous cell carcinoma[55]. We hypothesize that this result reflects that it is harder to recall the number of childhood sunburns when that number is large compared with having few or no sunburns, thus increased predisposition to sunburns will increase both IDK responses and skin cancer risk. This result suggests that our factor score GWAS successfully highlights shared components affecting item nonresponse generally ($r_g > 0$) while avoiding capturing more question-specific non-response behaviour that is less related to overall nonresponse (for example, $P = 0.326$ for rs35407).

## Genetic correlations with heritable traits

To better understand which traits and behaviours share genetic overlap with item non-response behaviour, we calculated genetic correlations between the PNA and IDK factors with 654 additional heritable phenotypes, 615 of which were internal to UKB and 39 were obtained from previously published GWAS, using Linkage Disequilibrium Score Regression (LDSC) (Supplementary Table 8). Note that 109 out of 654 (16.67%) traits tested were included in the set of questions used to derive the item non-response phenotypes.

Consistent with our phenotypic findings reported above, the PNA and IDK factors are strongly genetically correlated with other survey participation in UKB. Specifically, we observed a strong genetic correlation between our factors and agreeing to participate in the online follow-up diet questionnaire (PNA $r_g = -0.47 (0.05)$, $P = 6.75 \times 10^{-19}$; IDK $r_g = -0.29 (0.04)$, $P = 1.00 \times 10^{-15}$), indicating that associations with item-level nonresponse partially overlap with those of response to follow-up. The PNA and IDK factors also show high genetic correlation with opting to skip the sexual history section in the UKB survey (PNA $r_g = 0.58(0.04)$, $P = 1.61 \times 10^{-46}$; IDK $r_g = 0.50(0.03)$, $P = 1.05 \times 10^{-52}$). Importantly neither of these questions were used for deriving PNA and IDK factors, providing additional evidence that the derived phenotypes are indeed capturing consistent item non-response behaviour in the UK Biobank.

More broadly, we observed strong genetic correlations between the PNA and IDK factors and a wide range of phenotypes including health indicators (for example, $r_{g\,PNA} = 0.51(0.03)$ and $r_{g\,IDK} = 0.49(0.02)$ with overall health rating), psychiatric disorders (for example, $r_{g\,PNA} = 0.32(0.05)$ and $r_{g\,IDK} = 0.30(0.03)$ with attention-deficit

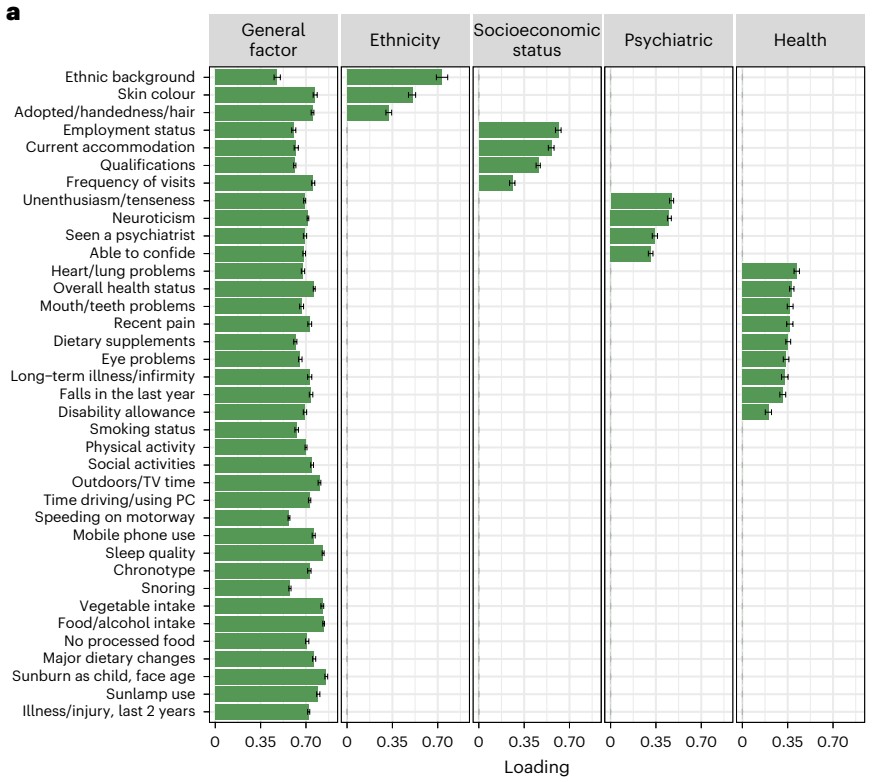

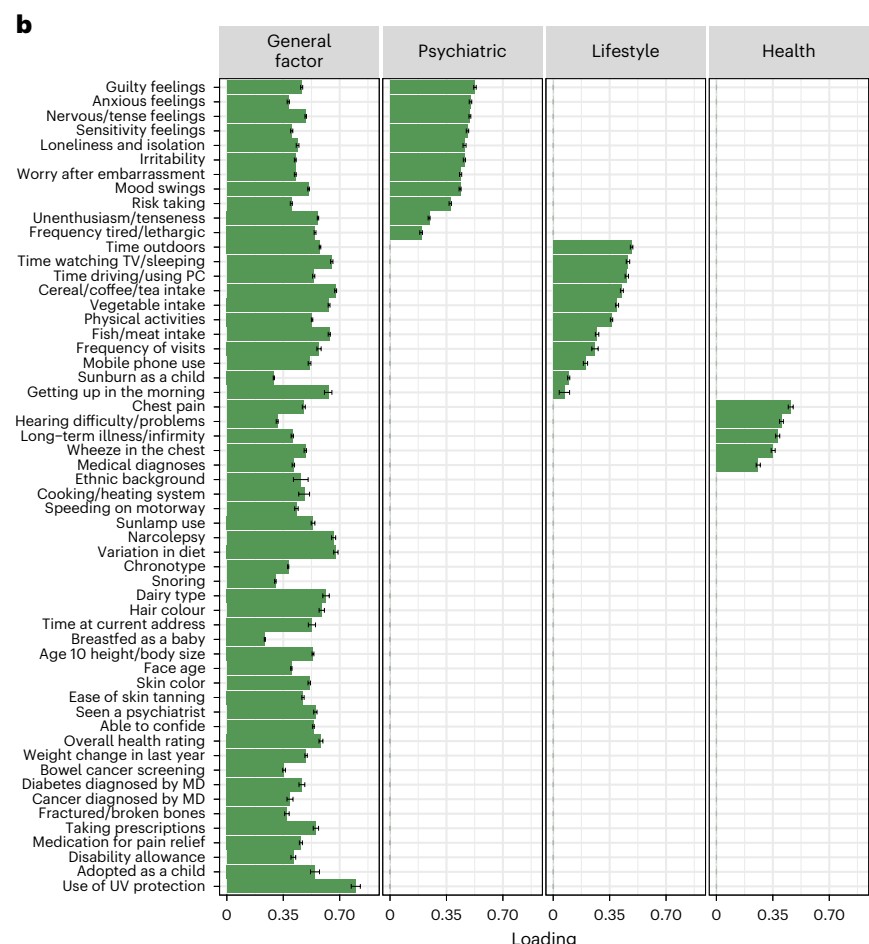

**Fig. 3 | Bar graph of factor loadings of the questions in the PNA and IDK confirmatory factor analyses.** For each item in the analysis: **a**, The fitted loadings of PNA responses on each latent factor in the CFA with the selected bifactor model for PNA. **b**, The fitted loadings in CFA of the bifactor model for IDK. Error bars indicate ±1 s.e. Loadings without bars were constrained to 0 in the corresponding CFA.

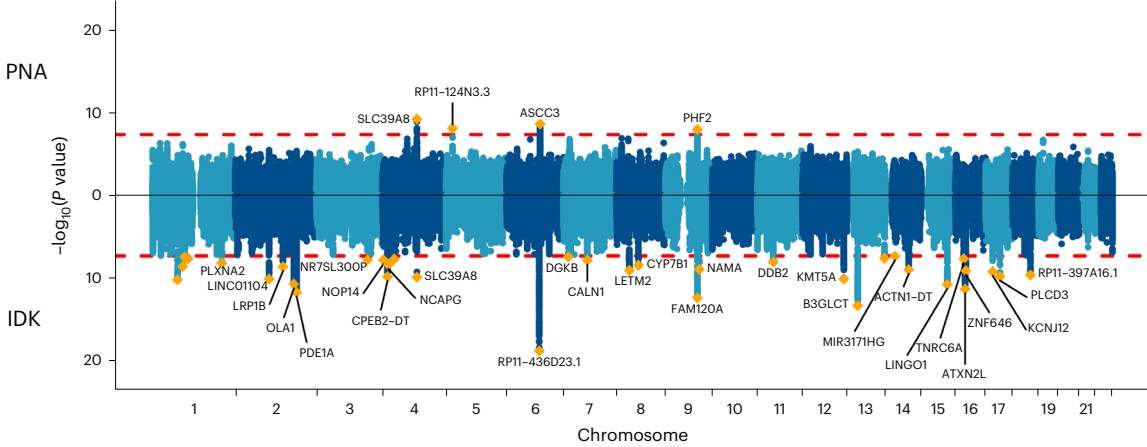

**Fig. 4 | Miami plot for GWAS of the PNA and IDK factors.** Miami plot of the resulting *P* values from a GWAS of the PNA factor (top) and the IDK factor (bottom) for the 360,628 UK Biobank participants included in this study. The *x* axis is the chromosomal position and the *y* axis represents the $-\log_{10}$ of the corresponding *P* value for each location measured in the genome with a two-sided test. The dashed red line marks the threshold for genome-wide significance ($P = 5 \times 10^{-8}$), which corrects for multiple hypothesis testing.

hyperactivity disorder), personality measures (for example, $r_{\text{g\_PNA}} = 0.34(0.03)$ and $r_{\text{g\_IDK}} = 0.24(0.02)$ with neuroticism score) and cognition (for example, $r_{\text{g\_PNA}} = -0.48(0.03)$ and $r_{\text{g\_IDK}} = -0.27(0.03)$) with fluid intelligence score (Fig. 5 and Supplementary Table 8). As with the pleiotropic top hits, this overlap with other GWAS can be expected to reflect a mix of truly shared genetic risk factors and inflated similarity to item nonresponse due to selection bias. Among the top genetic correlations for both non-response factors are traditional indicators of SES such as educational attainment ($r_{\text{g\_PNA}} = -0.51(0.03)$ and $r_{\text{g\_IDK}} = -0.38(0.02)$) and total household income before tax ($r_{\text{g\_PNA}} = -0.57(0.04)$ and $r_{\text{g\_IDK}} = -0.46(0.02)$). These results are consistent with previous work[33] suggesting that non-response behaviours are strongly linked with SES.

Given the strong previous evidence for the association of non-response with SES, we next evaluated the degree to which the overall pattern of genetic correlations could be explained by SES using genomic structural equation modelling[56,57] (genomic SEM). First, we estimated that total household income[47], region-based social deprivation[58] and education[46]—items traditionally considered major indicators of SES[47,59]—jointly explain 34% of the SNP heritability in PNA (standardized residual variance = 0.664(0.0815)) and 22% of the heritability in IDK (standardized residual variance = 0.782(0.0392)) (Supplementary Fig. 8a). We then estimated genetic correlations between PNA, IDK and the remaining 654 heritable phenotypes with the same control for income, local social deprivation and educational attainment GWAS with genomic SEM (Supplementary Fig. 8b). Overall, we observed a decrease in the number of traits significantly correlated with PNA and IDK factors after performing this analysis, suggesting that many of the observed genetic correlations for PNA and IDK may be at least partially explained by SES (Supplementary Fig. 9 and Table 9). On the other hand, both the PNA and IDK factors remained at least nominally associated with poor self-reported overall health ($r_{\text{g}} = 0.27$, $P = 2.46 \times 10^{-4}$ and $r_{\text{g}} = 0.30$, $P = 4.75 \times 10^{-13}$, respectively). The IDK factor in particular remained significantly associated with a number of health-related items, such as 'Number of self-reported non-cancer illnesses' ($r_{\text{g}} = 0.23$, $P = 2.93 \times 10^{-12}$) and 'Long-standing illness, disability or infirmity' ($r_{\text{g}} = 0.22$, $P = 3.00 \times 10^{-9}$), while PNA retained specific associations to psychiatric items such as schizophrenia ($r_{\text{g}} = 0.18$, $P = 4.92 \times 10^{-5}$) and feeling 'tense' or 'high-strung' ($r_{\text{g}} = 0.28$, $P = 4.87 \times 10^{-8}$). These results suggest that genetic association of nonresponse with poor overall physical and mental health is not fully accounted for by the genetics of socioeconomic factors. Overall, these results highlight traits that may

be genetically associated with nonresponse and are at highest risk of being affected by bias from nonrandom missingness, beyond what we can learn from using phenotypic information alone.

**Independent associations with PNA and IDK**

In addition to the genetic correlation of the PNA and IDK factors with other traits, we observe substantial genetic correlation between these two factors ($r_{\text{g}} = 0.73(0.03)$, $P = 3.92 \times 10^{-125}$), reflecting partial but not complete ($r_{\text{g}} < 1$, $P = 1.20 \times 10^{-19}$) genetic overlap between these two factors. Notably, genetic correlation facilitates this comparison between forms of item nonresponse by using genetics to overcome the limitation that UK Biobank participants could only respond with one of PNA or IDK on a given item, not both. The substantial genetic correlation may in part reflect shared aetiology or overlapping use of the nonresponse options beyond their literal meaning, such as reporting IDK to avoid invoking personal preference[41] or reporting PNA when an item's answer is not immediately obvious.

To help understand which genetic components are unique to PNA and IDK, we considered traits whose correlation with the two non-response types differed (Supplementary Fig. 8c). Of the 654 additional traits tested, 38 had significantly different genetic correlation estimates for PNA and IDK (Supplementary Table 8 and Fig. 5c). Among these 38 phenotypes, PNA had stronger genetic correlations with psychiatric (for example, schizophrenia[60] $r_{\text{g\_PNA}} = 0.21(0.03)$ vs $r_{\text{g\_IDK}} = -0.006(0.02)$, $P_{\text{diff}} = 3.72 \times 10^{-12}$), cognitive (for example, general cognitive performance[45] $r_{\text{g\_PNA}} = -0.46(0.03)$ vs $r_{\text{g\_IDK}} = -0.27(0.02)$, $P_{\text{diff}} = 3.33 \times 10^{-12}$) and sociodemographic variables (for example, educational attainment[46] $r_{\text{g\_PNA}} = -0.51(0.03)$ vs $r_{\text{g\_IDK}} = -0.38(0.02)$, $P_{\text{diff}} = 2.05 \times 10^{-8}$). IDK showed more substantial correlation with reported activities (for example, using UV protection $r_{\text{g\_IDK}} = -0.12(0.03)$ vs $r_{\text{g\_PNA}} = 0.05(0.02)$, $P_{\text{diff}} = 1.41 \times 10^{-6}$) and nutrition (salad intake $r_{\text{g\_IDK}} = -0.21(0.03)$ vs $r_{\text{g\_PNA}} = -0.02(0.04)$, $P_{\text{diff}} = 5.83 \times 10^{-7}$).

Given the strong genetic correlation between IDK and PNA, these differences create the possibility that the genetic correlation between one of the non-response types and an outside trait may be fully explained by correlations with the other non-response type. To test this possibility, we estimated the conditional genetic correlation between PNA and other heritable phenotypes controlling for the genetic associations of IDK and vice versa for correlations with IDK conditional on PNA, using genomic SEM[57] (Supplementary Fig. 8d). After accounting for the shared genetic associations between PNA and IDK, much of the genetic correlation observed between IDK

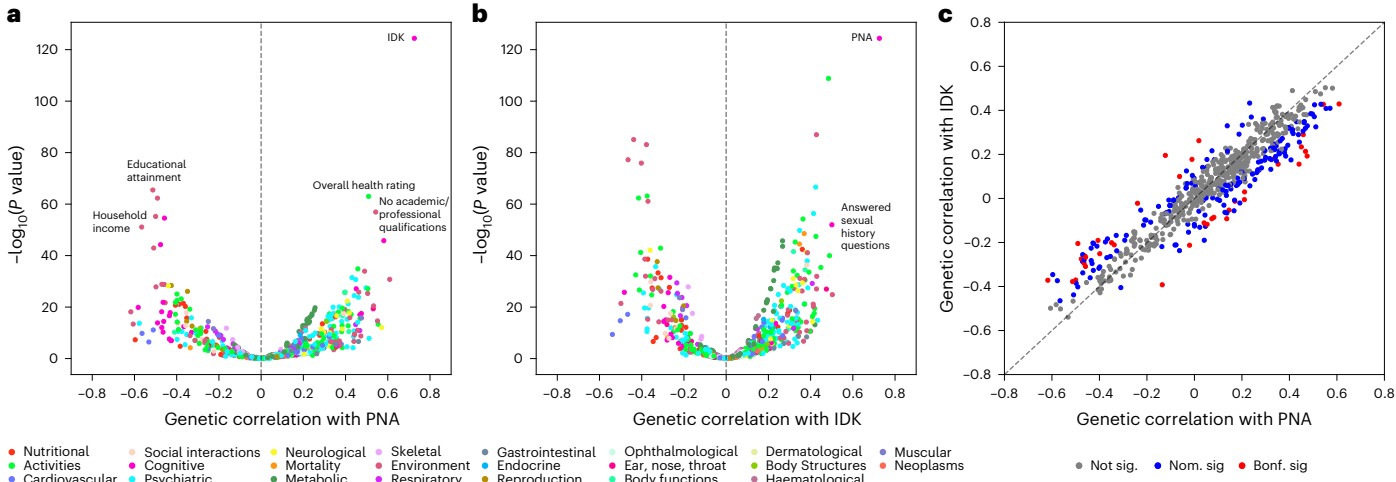

**Fig. 5 | Genetic correlations between the non-response factors and other heritable traits. a,b**, Genetic correlations between our PNA (**a**) and IDK (**b**) factors and 655 other traits. The values on the *x* axis are the point estimate of the genetic correlation. The values on the *y* axis represent $-\log_{10}$ of the *P* value of the associated test statistic. Only traits with a genetic correlation (in absolute value) >0.50 and with $P < 10^{-50}$ are labelled. **c**, Relationship between genetic correlation point estimates for other traits with PNA and IDK. Point estimates for PNA are shown on the *x* axis and those for IDK on the *y* axis. In **a** and **b**, marker colour reflects the category of the trait. In **c**, colour indicates the significance level (that is, not significant: $P > 0.05$, nominally significant: $P < 0.05$, significant after multiple-hypothesis Bonferroni correction for 654 traits tested: $P < 7.65 \times 10^{-5}$) for the difference between PNA and IDK correlations with a given trait. All statistical tests for **a**, **b** and **c** are two-sided.

and SES-related traits was reduced (Supplementary Fig. 10). The 'PNA-adjusted' IDK factor was no longer even nominally associated with educational attainment ($r_g = -0.01$, $P = 0.830$) and total household income ($r_g = -0.10$, $P = 0.136$), while the association with self-rated general health ($r_g = 0.20$, $P = 7.33 \times 10^{-4}$) was attenuated. Conversely, 'IDK-adjusted' PNA maintained significant associations with educational attainment ($r_g = -0.38$, $P = 3.45 \times 10^{-19}$), income ($r_g = -0.37$, $P = 2.11 \times 10^{-11}$) and general health ($r_g = 0.26$, $P = 1.28 \times 10^{-7}$). Corresponding analyses of conditional genetic correlations for psychiatric disorders yielded a cross-over effect, with 'PNA-adjusted' IDK correlations with schizophrenia ($r_g = -0.23$, $P = 3.58 \times 10^{-7}$) and bipolar disorder ($r_g = -0.21$, $P = 4.99 \times 10^{-8}$) moving towards the opposite sign of the observed genetic correlations with PNA. Taken together, these results highlight that although PNA and IDK have substantial genetic overlap, they remain partially distinct. The genetic association of IDK with education and other SES-related variables may be largely explained by genetic factors shared with PNA, while the association of PNA with psychiatric phenotypes appears to involve more genetically distinct elements.

**Polygenic score analysis of the National Longitudinal Study of Adolescent to Adult Health (Add Health) data**
To test the generalizability of our genetic findings, we constructed polygenic scores for the PNA and IDK factors in Wave 4 of the Add Health data. Item nonresponse in Add Health was identified on the basis of 163 questions with a possible response of 'I don't know' and 217 questions with a possible response of 'refused to answer' (Supplementary Table 10). The IDK and PNA polygenic scores showed significant association with whether individuals gave a corresponding IDK or PNA response to at least one question. Specifically, using logistic regression models, we estimated that a one-standard-deviation increase in the PNA polygenic score is associated with a 2% increase in the probability of an individual ever answering with 'refused to answer' in the Wave 4 Add Health data (incremental pseudo-$R^2 = 0.1\%$). We also estimated that a one-standard-deviation increase in the IDK polygenic score is associated with a 2% increase in the probability of an individual ever answering with 'I don't know' in the Wave 4 Add Health data (incremental pseudo-$R^2 = 0.5\%$). Taken together, these results suggest

that our findings in the UK Biobank replicate in an external US-based study of younger individuals.

We also considered whether better polygenic prediction can currently be achieved in Add Health using a recently developed polygenic score for educational attainment (continuous years of completed education)[46], theorizing that a score for a trait highly genetically correlated with nonresponse derived from a much larger sample size ($N = 1,131,881$) could improve prediction of our two non-response outcomes (Supplementary Table 11). Better prediction of nonresponse from a different polygenic score could be of interest for improving the power of some methods for correcting selection bias. We found that the educational attainment polygenic score was significantly associated with a 1% increase in the probability of an individual ever answering 'refused to answer' in the Wave 4 Add Health data (incremental pseudo-$R^2 = 0.08\%$) and was not significantly associated with the 'I don't know' outcome, suggesting that polygenic scores based on the GWAS of IDK and PNA should currently be preferred for prediction of overall nonresponse.

**Heckman correction of GWAS results**
Finally, we evaluate the potential impact of adjustment for missingness from nonresponse in GWAS using Heckman correction[61]. Briefly, the Heckman two-step estimator first builds a selection model for each individual's likelihood of missingness and then uses this prediction of missingness as a covariate to adjust the regression model of interest (for example, GWAS; Methods). As a proof of concept, we apply this approach to GWAS of FI measured at follow-up in UKB ($N = 83,677$), since it has substantial item- and survey-level missingness and strong genetic correlations with both PNA ($r_g = -0.39(0.04)$) and IDK ($r_g = -0.26(0.03)$). We find that the selection model modestly predicts missingness in FI (pseudo-$R^2 = 0.045$) with significant effects of the factor scores for PNA ($P = 1.46 \times 10^{-69}$) and IDK ($P = 1.93 \times 10^{-43}$). The predicted missingness is then strongly associated with FI in the response model (Heckman $\lambda$ coefficient $= 2.886(0.042)$, $P < 1 \times 10^{-308}$), suggesting that the observed missingness is informative and correction may reduce bias.

In this example application, we find substantial differences between GWAS results for FI with and without using Heckman

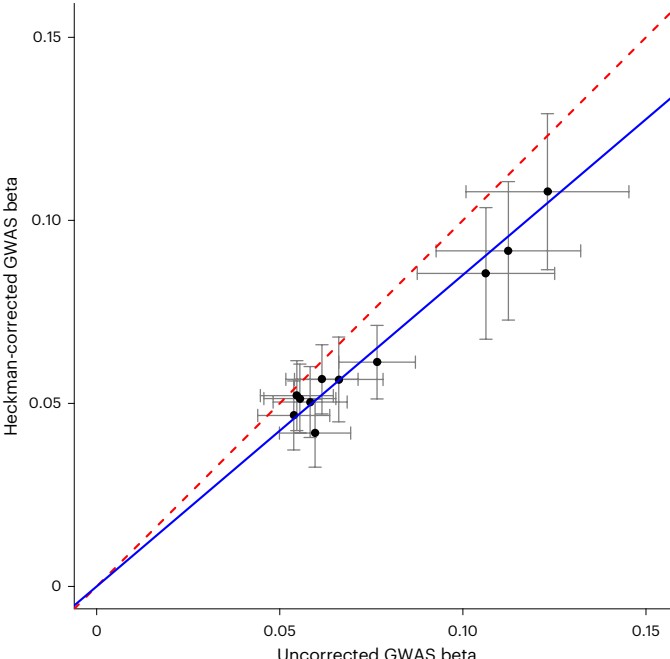

**Fig. 6 | GWAS results for lead SNPs associated with FI with and without Heckman correction for missingness in $N$ = 83,677 UKB participants in a follow-up FI survey.** Deming regression of the observed betas in the two GWAS (solid blue line) indicates that betas from the Heckman corrected GWAS are, on average, attenuated downward from being equal to the uncorrected GWAS (dashed red line). Grey error bars reflect ±1 s.e. on the estimated betas for each SNP. Lead SNPs for GWAS of FI identified using FUMA[88]. Deming regression performed with the R package 'deming'[101].

correction. Of 11 loci significantly associated with FI in the uncorrected GWAS, only 3 of the lead SNPs remain genome-wide significant after Heckman correction (Supplementary Table 12), with effect sizes of these lead SNPs being 15% smaller on average (Deming regression slope = 0.851(0.023), $P$ = 8.86 × 10⁻¹¹ for test of slope = 1; Fig. 6). Genome-wide, Heckman correction yields a small but significant difference in the overall pattern of genetic results ($r_g$ = 0.967(0.002), $P$ = 2.14 × 10⁻³⁹ for test of $r_g$ = 1). Encouragingly, the Heckman correction reduces genetic correlation with PNA ($r_g$ = −0.28(0.04)) and IDK ($r_g$ = −0.11(0.03)), with the remaining correlation possibly reflecting incomplete correction for nonresponse or true underlying correlation with non-response behaviour. More substantively, meaningful differences in genetic correlation with GWAS of other traits are observed, such as a much weaker correlation with coronary artery disease after Heckman correction (uncorrected GWAS $r_g$ = −0.14(0.03), corrected GWAS $r_g$ = −0.04(0.03); Supplementary Table 13). However, we strongly caution against adopting these proof-of-concept results as definitive for the genetics of FI, since they are sensitive to the specification of the selection model (Supplementary Information and Table 14) and additional modelling is probably appropriate to fully capture the relevant missingness mechanisms[62,63]. Taken together, these results broadly demonstrate the potential impact of correcting for nonresponse on substantive GWAS findings.

## Discussion

Nonresponse can impact the generalizability and reliability of survey-based research[64,65]. We show that overall item-level nonresponse is not random in the UK Biobank, with a significant heritable component. These findings on item-level nonresponse provide insight on more granular non-response behaviour to extend recent work characterizing the genetics of baseline study participation[66] and participation

at follow-up[33,38]. Given the critical importance of respecting the ethical boundaries presented by the individual's stated decision not to respond to a given item (Box 1), the current analysis focuses on characterizing overall item non-response behaviour, avoiding inference about item-specific reasons for nonresponse. We demonstrate that accounting for the selection bias induced by the overall likelihood of nonparticipation has the potential to substantially impact other GWAS results.

The current results for IDK and PNA suggest substantial similarity in the use of these two forms of nonresponse, but with some response-specific features. Phenotypically, both IDK and PNA exhibit substantial correlation across questions, both broadly and within clusters that reflect the survey's content and structure. Strong genetic correlation is observed between the GWAS of estimated general factors for PNA and IDK across items, but the correlation is significantly less than 1 ($r_g$ = 0.73(0.03)). Notably genetics enables this comparison despite the forced choice between responding either IDK or PNA to any given item in UKB. Among genome-wide significant loci, two of the four loci associated with PNA are also associated with IDK, and three of IDK-associated loci have recently been associated with at least one aspect of study participation (that is, providing valid recontact information[33], participation in follow-up surveys[38] and sex-differential participation bias[39]). These results are consistent with the idea that item nonresponse, unit nonresponse and study nonparticipation may be associated with a shared spectrum of factors, and this shared spectrum may have a genetic component, with remaining residual genetic factors more specific to particular forms of nonresponse.

Given that previous literature suggests a strong relationship of nonresponse with socioeconomic status and health[26–28,38,39], we carefully characterize the behaviour of our phenotypic and genetic findings with respect to these variables. First, we recapitulate the previously reported association with these variables, showing that both IDK and PNA also strongly genetically correlate with education and self-reported health. These results are consistent with recent findings for polygenic associations with attrition[37] and study participation[66]. Second, conditional genetic correlation analysis suggests that the genetic correlation of IDK with socioeconomic status (that is, educational attainment and household income) can be largely explained by its correlation with PNA. This may indicate that the genetic overlap between IDK and PNA includes the genetic correlates of SES, with additional SES-related effects potentially unique to PNA responses. Third, IDK and PNA continue to show significant genetic and phenotypic associations conditional on SES and health variables. This is important for two reasons: (1) it suggests that non-response behaviour is not fully mediated through, for example education, and thus associations beyond these variables need to be identified to fully characterize nonresponse and (2) the current estimates for IDK and PNA behaviour in this paper are already sufficient to improve prediction of nonresponse both phenotypically (that is, with estimated factor scores in UKB) and genetically (that is, with polygenic scores), including generalization of the polygenic score to other samples. As discussed below, accurate modelling of non-response behaviour is likely to be a key component of successfully correcting GWAS and other analysis for selection bias.

Observed genetic correlation of the nonresponse factors with other traits may provide additional insight into potentially unique genetic components of IDK and PNA. For instance, PNA appears to have a unique genetic correlation with psychiatric phenotypes (for example, schizophrenia) not observed in IDK, while IDK shows some signs of unique correlation with health-related diet and lifestyle behaviours. If we presuppose that these correlations reflect true shared genetic variants associated with nonresponse (for example, path $\alpha \times \theta$ in Fig. 1), then these results may for instance suggest that PNA partially reflects nonresponse more related to anxiety over the question (for example, 'tense'/'high-strung' (UKB code 1990) $r_{g,IDK}$ = 0.183(0.025), $r_{g,PNA}$ = 0.339(0.041)), while some IDK responses may reflect strategic

avoidance of disclosing some health-related behaviours (for example, salad/vegetable intake (UKB code 1299) $r_{g,IDK} = -0.213(0.031)$, $r_{g,PNA} = -0.021(0,045)$). The latter hypothesis would be consistent with the possibility that the higher phenotypic rate of IDK responses is associated with social desirability bias or satisficing[41]. On the other hand, presupposing that the observed genetic correlation reflects collider bias (for example, paths $\theta$ and $\beta > 0$ in Fig. 1) would invert these interpretations. In either case, these correlations are expected to reflect the combination of effects of selection bias and shared genetic associations with phenotypic drivers of nonresponse. While we were unable to fully explore this question in this study, future work might focus on additionally triangulating these comparisons with GWAS of other elements of participation and nonresponse, and how they generalize to nonparticipation in other cohorts to better distinguish between evidence of uncorrected selection bias in existing GWAS and true associations with nonresponse.

Finally, this investigation of the genetics of nonresponse in UKB is motivated primarily by the desire to understand and address selection bias in other GWAS. The current results demonstrate that simple correction (that is, the two-step Heckman estimator) of GWAS using the current estimates of general IDK and PNA behaviour has the potential to substantially impact observed genetic results. Our Heckman correction however has a key limitation: the effectiveness of this correction will depend on having the ability to correctly specify an accurate model for the missing data mechanisms affecting the sample[67]. The same is probably true for the many other possible correction methods that have been proposed (for example, full information maximum likelihood[68], imputation[40], sample weighting[14,69], instrumental variables[62] or matching to population demographics[18]). Identifying genetic associations with missingness, whether from item nonresponse or other selection effects, has the advantage of aiding this modelling of missingness solely from genetic data without advanced knowledge of what phenotypic measures explain the missingness.

Where possible, however, these genetic elements should be supplemented by considering phenotypic correlates of nonresponse within the sample[70] and/or comparison to expected population descriptive statistics from national registries[16] or by linkage to individual data[71], since those analyses will still provide valuable information about participation and nonresponse, and highlight what individuals may be underrepresented in the study or particular analysis. While we were limited in our correction exercise here, we anticipate that ideal correction of selection bias in GWAS will require incorporating the associations with item nonresponse identified here with findings on unit nonresponse[33,38], study participation[66] and other ascertainment effects[16,39], as well as other confounds such as stratification and longitudinal change or misreporting[72].

In conclusion, we use phenotypic and genetic data to provide an investigation of overall item-level nonresponse across items in the UK Biobank. These results should be considered when analysing the UK Biobank, among other biobank-scale survey efforts, and when developing novel methods aimed at correcting and leveraging nonresponse in genetic analyses. We also encourage readers to carefully consider the ethical considerations and interpretations of our work, which are highlighted in Box 1, Methods and the Supplementary Information.

## Methods

### Ethical approval
Use of the UKB data was approved under application 31063. Because of the sensitive nature of this study, we also explicitly sought permission for the specific scope of this paper (that is, 'to also study response rates and response characteristics (for example, how often a response is left unanswered) and to examine whether there are any genetic factors that correlate with these response phenotypes'), which the UK Biobank granted under the same application.

Analysis of the UKB data was reviewed by the Partners HealthCare IRB (Partners Human Research), which determined in expedited review that the project met the US federal criteria definition of 'not human subjects research' (Protocol no. 2019P000883 titled 'Behavioural Genetics Study of Responsiveness from UKBB Questionnaires').

Analysis of the Add Health data was reviewed by the Office of Research Subject Protection (OSRP) at the Broad Institute of MIT and Harvard, which determined that the project met US federal criteria for exemption from IRB review (Project no. 0001 titled 'Genetic and environmental factors influencing complex social behaviour').

### UKB and inclusion criteria
The UKB is a health resource which has the purpose of improving the prevention, diagnosis and treatment of human disease[73]. It consists of a prospective cohort of 502,620 men and women aged 40–69 recruited in the years 2006–2010 throughout the United Kingdom. The touchscreen questionnaire is a collection of self-reported information regarding general health, dietary habits, physical activity, psychological and cognitive states, sociodemographic factors and so on. We began with 361,194 unrelated individuals of European genetic ancestry who passed quality control measures (https://www.nealelab.is/uk-biobank/ukbround2announcement). We excluded individuals who were enrolled only in the UKB pilot study ($N = 335$). Participants who decided to terminate the touchscreen questionnaire were asked to select PNA to all subsequent questions and they were kept in our analyses. Conversely, individuals who withdrew from the study without filling out the touchscreen survey were excluded from the analysis ($N = 231$). As a result, a total of $N = 360,628$ participants took part in the survey and answered every question of interest in the study; this is the final analytic sample size.

### UKB item non-response definitions
We considered only the touchscreen questionnaire phenotypes with the response options 'Prefer not to answer' (PNA, response code -3) or 'I don't know' (IDK, response code -1). Items were included in our analyses only if valid responses existed for all participants in our sample ($N$ questions = 109 and 83 for PNA and IDK, respectively). This included one instance of a derived item that retained nonresponse information (UKB FieldID 20116). Items excluded from our analyses thus included those with incomplete data due to, for example, being asked only to a subset of participants, or being added to the touchscreen questionnaire later in the recruitment process.

### National Longitudinal Study of Adolescent to Adult Health (Add Health) cohort
Add Health originated as an in-school survey of a nationally representative sample of US adolescents enrolled in grades 7 through 12 during the 1994–1995 school year[74]. Respondents were born between 1974 and 1983, and a subset of the original Add Health respondents has been followed up with in-home interviews, which allows researchers to assess correlates of outcomes in the transition to early adulthood. In Add Health, the mean birth year of respondents is 1979 (s.d. = 1.8) and the mean age at the time of assessment (Wave 4) is 29.0 years (s.d. = 1.8). All phenotypes included in this study came from Wave 4, the latest wave of Add Health data collection (2007–2009).

### Phenotype definitions in Add Health
To investigate item non-response bias phenotypes in Add Health, we considered two possible answer choices across hundreds of questions from the Wave 4 Add Health in-home interview questionnaire: 'refused to answer' and 'I don't know'. The final study population included 3,414 unrelated participants of European genetic ancestry with available genetic data. The 'refused to answer' option was available for 217 questions, while only 163 questions allowed the 'I don't know' option. Our final outcomes were whether respondents ever answered at least once with 'refused

to answer' or 'I don't know', respectively. We also predicted the two non-response outcomes in Add Health using a recently developed polygenic score for educational attainment[46] (completed years of education).

### Factor model construction

**Single-factor model.** Exploratory factor analysis (EFA) with a single latent factor was performed separately on tetrachoric correlation matrices between each of the dichotomized PNA and IDK responses and was implemented using the fa function of the psych package in R software (v.3.4.4) with the oblique rotation 'biquartimin' and the 'ordinary least squares' extraction method.

Residuals from the initial EFA revealed a correlation structure indicative of further clustering unaccounted for by the general factor, with both broad correlations across item domains as well as some highly specific pairwise structure at the item level. Given that we were interested in modelling overall non-response behaviour, not behaviour specific to or driven by single item groupings or domains, we sought to reduce this additional structure first by pruning items with highly correlated non-response patterns and then by fitting a bifactor model. This initial pruning step was necessary to flatten the nested correlation structure observed within domains to facilitate fitting of the more interpretable bifactor model. To implement this pruning, we performed agglomerative clustering of the residuals from the single-factor EFA and inspected the resulting dendrogram (Supplementary Figs. 3 and 4). We chose to cut the dendrograms at height 0.500 and 0.775 in PNA and IDK, respectively, to minimize the number of branches (that is, clusters of variables grouped together) while also maintaining homogeneity within these branches (for example, questions belonging to the same field). This led to 37 and 56 branches in PNA and IDK, respectively. In the IDK analysis, summing the IDK for each participant across questions inside each branch was sufficient to reduce the substructure of the residuals. For PNA, we observed that in the four largest branches (that is, person-specific information, food intake, overall health status and mental health), the distribution of PNA per participant was J-shaped, with a large number of individuals with zero PNA responses in the branch, a continuously decreasing number of participants who chose PNA in 1 or more question and a small peak of individuals who chose PNA in every question in the branch. For this reason, we scored each participant as follows: '0' if a participant answered all questions (that is, never chose PNA), '1' if a participant chose only PNA only once, '3' if a participant preferred not to answer all the items that fell in the same branch and '2' for participants who did not fit into the previous three categories. These scores were input as ordinal values for bifactor analysis, allowing for minimal item non-response information loss while limiting the influence of individuals responding PNA to all questions.

**Bifactor model.** The bifactor model is a factor analysis model that assumes the observed covariance between items (approximately) reflects the contributions on one shared factor influencing all items and two or more specific factors influencing subsets of items[75,76]. This model was chosen to reflect the apparent structure of correlations among the observed item nonresponse with a general factor for overall response behaviour (IDK or PNA) and specific factors influencing sets of items with related content. We chose to focus on the results for this general factor since we expect it to be more generalizable to nonresponse on a broad range of items in UKB and other studies, and to be less influenced by the specific set of questions included in our analyses.

To run bifactor analysis on the pruned set of UKB questions, we first split the dataset between 80% of participants ($N = 288,502$) for EFA[77] and 20% of participants ($N = 72,126$) for CFA. For EFA, we used the fa function, with 'biquartimin' and 'OLS' as the rotation and factoring methods, respectively. We implemented CFA using the cfa function from the lavaan package[78] in R software (v.3.4.4) and also using the weighted least square mean and variance adjusted estimator. We selected the initial factor structure from the EFA, first fitting models

with different numbers of domain factors (Supplementary Table 3), then confirmed the fit of the model in the hold-out sample using the root mean square error of approximation and the Tucker–Lewis Index. Upon selecting the optimal model and confirming fit, we re-ran the CFA in the full combined dataset; the final PNA and IDK phenotypes used in all downstream analyses were obtained as factor scores of the CFA-derived general factor in the full dataset. We extracted the factor scores using the 'Empirical Bayes Model' method as implemented in the lavPredict function.

**Predicting participation and item-level nonresponse in follow-up questionnaires.** We ran logistic regression to predict item nonresponse on a follow-up mental health questionnaire using our PNA and IDK factor scores. Of 141 total items on the mental health follow-up questionnaire, only 58 contained valid responses for all participants ($N = 118,037$ in our sample) and, of those, all had an option to respond PNA, while only 14 had an IDK option. Due to this imbalance and the hypothesis of shared overlap in PNA and IDK behaviours, we chose to collapse PNA and IDK responses across items, identifying 36,517 individuals (30.5%) responding either PNA or IDK at least once across the 58 included items. Logistic regression was performed to associate this binarized outcome with PNA and IDK factor scores computed at baseline, along with chromosomal sex, age, age$^2$, sex × age$^2$ and the first 20 principal components of the variance–covariance matrix of the genetic data as covariates. To determine the significance and magnitude of prediction, we calculated the incremental pseudo-$R^2$ (Cox-Snell, as implemented in the Python package statsmodels) of including PNA and IDK scores over the baseline covariates described above. We also examined the significance of the individual PNA and IDK coefficients. We repeated these analyses using the additional covariates college completion (UKB fieldID 6138) and self-reported overall health (UKB fieldID 2178) to determine whether associations with PNA and IDK remained conditional on these well-known predictors of nonresponse.

We additionally ran logistic regression to predict completion of an online follow-up 24 h recall dietary questionnaire (UKB fieldID 110001) by using our PNA and IDK factors as predictor variables. To measure the variance explained by the model, we computed the pseudo-$R^2$ using the McKelvey and Zavoina statistical method[79]. Completion of the first wave of a dietary questionnaire was coded as 1 if a participant completed this wave and 0 if a participant did not ($N = 69,735$ and $N = 146,712$, respectively). Individuals with missing values (NA), reflecting individuals not invited to the dietary follow-up, were removed from the analysis ($N = 144,181$). Similarly, completion of all 4 waves of the dietary questionnaire was coded as 1 if someone completed all 4 waves and 0 if someone did not complete all 4 waves of the questionnaire ($N = 19,097$ and $N = 99,151$, respectively). Participants coded as missing or who completed some but not all of the dietary questionnaires were not considered in this analysis ($N = 242,380$). We examined the association of our standardized factors with sex, age, age$^2$, sex × age$^2$, the first 20 principal components of the variance–covariance matrix of the genetic data, self-reported health (UKB fieldID 2178) and years of education. Years of education was created by recoding the Qualifications field (UKB fieldID 6138) as follows, according to the International Standard Classification of Education (ISCED) codes[46]:

(1) College or University degree (ISCED) = 20 yr of education
(2) Advanced (A) levels/Advanced Subsidiary (AS) levels or equivalent (ISCED 3) = 15 yr of education
(3) Ordinary (O) levels/General Certificate of Secondary Education (GCSE) or equivalent (ISCED 2) = 13 yr of education
(4) Certificate of Secondary Education (CSE) or equivalent (ISCED 2) = 12 yr of education
(5) National Vocational Qualification (NVQ) or Higher National Diploma (HND) or Higher National Certificate (HNC) or equivalent (ISCED 5) = 19 yr of education

(6) Other professional qualification (for example, nursing, teaching) (ISCED 4) = 17 yr of education
(7) None of the above (ISCED 1) = 6 yr of education

Participants who chose PNA or IDK for either years of education or self-reported health were excluded.

**Genotyping and imputation.** Genotyping and imputation procedures for UKB have been detailed previously[73]. Briefly, UKB participants were genotyped using the Affymetrix UK BiLEVE Axiom array and UKB Axiom array. After extensive quality control, imputation was performed using SHAPEIT3[80] and a reimplementation of IMPUTE2[81] with the Haplotype Reference Consortium and merged 1000 Genomes[82] + UK10K[83] reference panels.

Genotyping in Add Health was performed at the Institute for Behavioral Genetics in Boulder, Colorado, using Illumina's Human Omni1-Quad-BeadChip[84]. After imputing the genetic data to the Haplotype Reference Consortium[85] using the Michigan Imputation Server[86], only HapMap3 variants were included, as these are well imputed and provide good coverage of common variation across the genome. Analyses were limited to individuals of European ancestry, and cryptically related individuals and ancestry outliers were dropped from analyses. Finally, only HapMap3 variants with a call rate above 98% and a minor allele frequency >1% were used.

**UKB GWAS.** We performed GWAS using linear regression implemented in Hail[87], including chromosomal sex, age, age$^2$, sex × age$^2$ and the first 20 principal components of the variance–covariance matrix of the genetic data as covariates. We included autosomal variants with imputation INFO column values > 0.8, minor allele frequency (MAF) > 0.01 and Hardy–Weinberg equilibrium (HWE) $P > 1 \times 10^{-10}$, as well as annotated protein-truncating or missense variants with MAF $> 1 \times 10^{-6}$ (based on Ensembl VEP consequence annotations). Following these filters, 9,367,367 total variants were included in the GWAS.

**Identification of independent loci.** We used the FUMA[88] pipeline to identify independent genomic loci. We considered an independent locus as the region including all SNPs in pairwise linkage disequilibrium ($r^2 > 0.6$), with the lead SNPs in a range of 250 kb and independent from other loci at $r^2 < 0.1$. We used the 1000 Genomes Phase3 Northern Europeans LD reference panel[82].

**Heritability and tissue-specific enrichment.** We used stratified LD Score regression (S-LDSR)[89] to estimate the proportion of variation in our PNA and IDK factors that is explained by inherited SNPs, using the Baseline v.1 model with 53 variant annotations for functional categories to better fit variability in effect sizes across the genome. For these analyses, as well as for all analyses involving LDSR, we filtered to SNPs in the HapMap3 reference panel that had MAF > 0.01 and INFO > 0.90 in our sample ($N = 1,089,172$).

S-LDSR[90] was also used to estimate heritability enrichment for certain regions of the genome on the basis of outside annotations conditional on the baseline model. For these analyses, we used three separate annotation sets based on previous papers[52]: (1) 10 different cell type groups based on unions of 220 individual histone-based annotations[43,90–94], (2) multitissue expression annotations ($N = 205$) based on GTEx and Franke lab data[52,53,95,96] and (3) annotation of differential gene expression across 13 different brain regions based in GTEx data[53]. Given the overlap between the annotations of brain tissues and their differential expression, multiple testing correction was only performed within-analysis (for example, $P < 0.005$ for the first annotation set using IDK summary statistics).

**Genetic correlation.** Genetic correlation between traits was computed using GWAS summary statistics with LD Score regression[97] using

reference panel LD estimates from European-ancestry individuals in the 1000 Genomes Project[82]. We ran genetic correlations for PNA and IDK with a total of 654 traits, 615 which were from UKB and are publicly available[98]. An additional 39 traits were selected from previous GWAS and spanned the domains of cognition, psychiatry, personality, medical diagnoses, physical characteristics and sociodemographics (Supplementary Table 8). Traits used for genetic correlation analyses were chosen before conducting the analyses, with the agreement of the coauthors.

**Genomic SEM.** Genomic SEM[56] is a two-stage structural equation modelling approach that operates on genetic, rather than phenotypic, covariance matrix. In the first stage, the genetic and sampling covariance matrices are estimated using LDSR. In the second stage, a multivariate system of covariance associations involving the genetic components of phenotypes are specified, and their corresponding parameters are estimated by minimizing the discrepancy between the model-implied covariance matrix and the empirical covariance matrix. We used genomic SEM to implement the following analyses: (1) obtain PNA and IDK residual heritability conditional on the genetic contributions of SES variables (Supplementary Fig. 8a), (2) adjust genetic correlations for the contributions of SES variables (Supplementary Fig. 8b), (3) determine the significance of differences in estimated genetic correlations of PNA and IDK with additional traits (Supplementary Fig. 8c) and (4) adjust genetic correlations of PNA and IDK with additional traits for the genetic contribution of the other nonresponse factor (that is, IDK and PNA, respectively; Supplementary Fig. 8d). For the purpose of these models, SES variables included total household income[47], region-based social deprivation[58] and educational attainment[46].

**Polygenic scoring.** Polygenic scores were constructed with LDpred[99]. LDpred estimates polygenic scores using SNP weights that estimate the conditional association of each SNP account for LD and the estimated genetic architecture of the trait, and has been shown to have greater prediction accuracy than conventional LD pruning followed by P value thresholding. We used a Wald test to evaluate the significance of the polygenic scores on the outcomes.

For the Add Health sample, we used the genotyped data from the Add Health prediction cohort to create the LD reference file. After imputing the genetic data to the Haplotype Reference Consortium[85] using the Michigan Imputation Server[86], we used only HapMap3 variants with a call rate >98% and a minor allele frequency >1% to construct the polygenic scores. We limited the analyses to European-ancestry individuals. Polygenic scores were calculated with an expected fraction of causal genetic markers set at 100%. In total, we used 1,168,025 HapMap3 variants to construct the polygenic scores in Add Health. We then used Plink[100] to multiply the genotype probability of each variant by the corresponding LDpred posterior mean over all variants. In total, we created two polygenic scores using the summary statistics of our two main phenotypes: PNA and IDK. We then determined the association of the polygenic score for the related 'refused to answer' and IDK phenotypes in Add Health. Prediction accuracy was based on a logistic regression of the outcome phenotype on the polygenic score and a set of standard controls, which included birth year, sex, an interaction between birth year and sex, and the first 10 genetic principal components of the variance–covariance matrix of the genetic data. Variance explained by the polygenic scores was calculated in regression analyses as the Nagelkerke's pseudo-$R^2$ change, that is, the pseudo-$R^2$ of the model including polygenic scores and covariates minus the pseudo-$R^2$ of the model including only covariates. The 95% confidence intervals around all pseudo-$R^2$ values were bootstrapped with 1,000 repetitions each. We also used a recently developed score for educational attainment[46] to predict both of our binary non-response outcomes in Add Health.

**Heckman correction.** To provide proof of concept for using knowledge of non-response patterns to correct associations for bias from

nonresponse, we demonstrated the impact of Heckman correction on GWAS associations. The Heckman two-step estimator (heckit)[61] first fits a selection model for missingness in the dependent variable using probit regression, for example:

$$E[I_{\mathrm{miss}}(y_i)] = \Phi\left(\sum_j \gamma_j z_{ij}\right) \qquad (1)$$

where $I_{\mathrm{miss}}(y_i)$ is an indicator function for missingness in $y_i$, and $\Phi()$ is the cumulative density function of the standard normal distribution. The response model of interest (that is, GWAS) then includes as a covariate the inverse Mills ratio of the predicted missingness liability $\lambda = \phi(\sum \gamma_j z_{ij})/\Phi(-\sum \gamma_j z_{ij})$, where $\phi()$ is the standard normal density function, which helps reduce bias by conditioning the expected value of the observed phenotype on the likelihood of selection. For the selection model, we included both nonresponse factors (that is, PNA and IDK), missingness-related SES and health variables (that is, educational attainment (UKB fieldID 6138, response code 1-College/University degree), total household income (UKB fieldID 738), neighbourhood Townsend social deprivation index (UKB fieldID 189) and self-rated health (UKB fieldID 2178)), along with chromosomal sex, age, age², sex × age², the first 20 principal components of the variance–covariance matrix of the genetic data and dummy codes for the baseline assessment centre. The response model included covariates for chromosomal sex, age, age², sex × age² and the first 20 principal components of the variance–covariance matrix of the genetic data. GWAS of the response model was implemented in Hail (https://github.com/hail-is/hail).

We demonstrated the effect of Heckman correction on associations with FI score in the UKB mental health follow-up survey (UKB fieldID 20191). We anticipated this variable to be a strong proof-of-concept given the strong observed genetic correlation of our non-response factors with fluid intelligence, its measurement after the items contributing to our non-response factor scores and the substantial informative missingness in the mental health follow-up[38].

**Reporting summary**

Further information on research design is available in the Nature Portfolio Reporting Summary linked to this article.

## Data availability

The GWAS results for our PNA and IDK phenotypes are available through GWAS catalogue accession nos. GCST90266936 for PNA and GCST90266935 for IDK. UK Biobank data are available to researchers via application at the following link: https://www.ukbiobank.ac.uk/enable-your-research/apply-for-access. Information about accessing 1000 Genomes Project or Hapmap3 data can be found at https://www.internationalgenome.org. For information about access to the data from this study, contact addhealth@unc.edu.

## Code availability

All software used to perform the analyses are available online. Scripts used to perform the analyses are available at https://github.com/gianmarcomigno/Item-nonresponse.

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

## Acknowledgements

R.W.'s work is supported by AnalytiXIN, which is primarily funded through the Lilly Endowment, IU Health and Eli Lilly and Company. B.M.N.'s work is supported by the Novo Nordisk Foundation (NNF21SA0072102). R.K.W. is supported by the Stanley Center for Psychiatric Research and R01 MH101244. This research was conducted by using the UK Biobank Resource under application 31063. A.G. was supported by the Academy of Finland (grant no. 323116) and by the European Research Council (ERC) under the European Union's Horizon 2020 research and innovation programme (grant no. 945733). This project also received funding from the European Union's Horizon 2020 research and innovation programme under grant agreement no. 101016775. The National Longitudinal Study of Adolescent to Adult Health (Add Health) is supported by grant P01 HD031921 to Kathleen Mullan Harris from the Eunice Kennedy Shriver National Institute of Child Health and Human Development (NICHD), with cooperative funding from 23 other federal agencies and foundations. Add Health GWAS data were funded by NICHD grants to Harris (R01 HD073342) and to Harris, Boardman and McQueen (R01 HD060726). The funders had no role in study design, data collection and analysis, decision to publish or preparation of the manuscript. We especially acknowledge and thank the participants in these studies for providing biological data and for their responses and nonresponses to survey questions that made this study possible.

## Author contributions

R.W., A.G. and R.K.W. designed and oversaw the study. G.M. was the study's lead analyst, responsible for quality control, meta-analyses and all major statistical analyses, under the supervision of C.E.C. Analysts who assisted G.M. in major ways include: R.W. (prediction), N.B. (GWAS analyses), M.C. and R.B. (statistical analyses). R.W. was responsible for the ethical oversight of the project and was assisted especially by K.F.M., who created the Frequently Asked Questions (FAQ) document. N.P., M.G.N. and B.M.N. provided helpful advice and feedback on various aspects of the study design and analyses. All authors contributed to and critically reviewed the manuscript. G.M., C.E.C., R.W., R.K.W. and A.G. made especially major contributions to the writing and editing.

## Funding

## Competing interests

C.E.C. is currently an employee of Novartis, although all work for this project was completed while she was a postdoctoral research fellow at the Broad Institute of MIT and Harvard. R.W. is a research fellow at AnalytiXIN, which is a consortium of health-data organizations, industry partners and university partners in Indiana primarily funded through the Lilly Endowment, IU Health and Eli Lilly and Company. B.M.N. is a member of the scientific advisory board at Deep Genomics and Neumora, and a consultant of the scientific advisory board of Camp4 Therapeutics. R.K.W. has received honoraria from the Jackson Laboratory and sponsored travel from the Russell Sage Foundation in the last 36 months. The remaining authors declare no competing interests.

## Additional information

**Correspondence and requests for materials** should be addressed to Robbee Wedow or Andrea Ganna.

[1]Analytic and Translational Genetics Unit, Massachusetts General Hospital and Harvard Medical School, Boston, MA, USA. [2]Institute for Molecular Medicine Finland, University of Helsinki, Helsinki, Finland. [3]Department of Statistics and Quantitative Methods, University of Milano-Bicocca, Milan, Italy. [4]Stanley Center for Psychiatric Research, Broad Institute of MIT and Harvard, Cambridge, MA, USA. [5]Center for Genomic Medicine, Massachusetts General Hospital, Boston, MA, USA. [6]Department of Sociology, Purdue University, West Lafayette, IN, USA. [7]Department of Medical and Molecular Genetics, Indiana University School of Medicine, Indianapolis, IN, USA. [8]AnalytiXIN (Analytics Indiana), Indianapolis, IN, USA. [9]Department of Statistics, Purdue University, West Lafayette, IN, USA. [10]Centre for Global Health Research, Usher Institute, University of Edinburgh, Edinburgh, Scotland. [11]Fondazione Human Technopole, Viale Rita Levi-Montalcini, Milan, Italy. [12]Department of Medical Epidemiology and Biostatistics, Karolinska Institutet, Stockholm, Sweden. [13]Department of Public Health, Purdue University, West Lafayette, IN, USA. [14]Department of Biological Psychiatry, Faculty of Behavioural and Movement Sciences, Vrije Universiteit, Amsterdam, the Netherlands. [15]Methodology Program, Amsterdam Public Health, Amsterdam, the Netherlands. [16]Amsterdam Neuroscience - Mood, Anxiety, Psychosis, Stress and Sleep, Amsterdam, the Netherlands. [17]Program in Medical and Population Genetics, Broad Institute of MIT and Harvard, Cambridge, MA, USA. [18]Novo Nordisk Foundation for Genomic Mechanisms of Disease, Broad Institute of MIT and Harvard, Cambridge, MA, USA. [19]These authors contributed equally: Gianmarco Mignogna, Caitlin E. Carey, Robbee Wedow. [20]These authors jointly supervised this work: Raymond K. Walters, Andrea Ganna. ✉e-mail: rwedow@purdue.edu; andrea.ganna@helsinki.fi

# Reporting Summary

## Statistics

For all statistical analyses, confirm that the following items are present in the figure legend, table legend, main text, or Methods section.

| n/a | Confirmed | |
|---|---|---|
| ☐ | ☒ | The exact sample size (*n*) for each experimental group/condition, given as a discrete number and unit of measurement |
| ☐ | ☒ | A statement on whether measurements were taken from distinct samples or whether the same sample was measured repeatedly |
| ☐ | ☒ | The statistical test(s) used AND whether they are one- or two-sided<br>*Only common tests should be described solely by name; describe more complex techniques in the Methods section.* |
| ☐ | ☒ | A description of all covariates tested |
| ☐ | ☒ | A description of any assumptions or corrections, such as tests of normality and adjustment for multiple comparisons |
| ☐ | ☒ | A full description of the statistical parameters including central tendency (e.g. means) or other basic estimates (e.g. regression coefficient) AND variation (e.g. standard deviation) or associated estimates of uncertainty (e.g. confidence intervals) |
| ☐ | ☒ | For null hypothesis testing, the test statistic (e.g. $F$, $t$, $r$) with confidence intervals, effect sizes, degrees of freedom and $P$ value noted<br>*Give P values as exact values whenever suitable.* |
| ☒ | ☐ | For Bayesian analysis, information on the choice of priors and Markov chain Monte Carlo settings |
| ☒ | ☐ | For hierarchical and complex designs, identification of the appropriate level for tests and full reporting of outcomes |
| ☐ | ☒ | Estimates of effect sizes (e.g. Cohen's *d*, Pearson's *r*), indicating how they were calculated |

*Our web collection on statistics for biologists contains articles on many of the points above.*

## Software and code

Policy information about availability of computer code

| | |
|---|---|
| Data collection | Data consisted of two primary nonresponse options "I Don't Know" and "Prefer Not to Answer" along with genetic data in the form of genotypes. Details on how these response options were extracted and analyzed are available in the manuscript and Methods. Details on how genotypes were obtained in each dataset are available in the Methods section of the manuscript. |
| Data analysis | Data analysis was conducted as specified in the Methods section of the manuscript. Hail Version 2.0 was used for GWAS, imputations was performed with IMPUTE2, factor analyses were performed with relevant packages in R Version 4.2.3, and polygenic scores were generated with PLINK Version 1.9 and LDpred2. All code has been updated and is available at https://github.com/gianmarcomigno/Item-nonresponse. |

For manuscripts utilizing custom algorithms or software that are central to the research but not yet described in published literature, software must be made available to editors and reviewers. We strongly encourage code deposition in a community repository (e.g. GitHub). See the Nature Portfolio guidelines for submitting code & software for further information.

## Data

Policy information about availability of data

All manuscripts must include a data availability statement. This statement should provide the following information, where applicable:
- Accession codes, unique identifiers, or web links for publicly available datasets
- A description of any restrictions on data availability
- For clinical datasets or third party data, please ensure that the statement adheres to our policy

> The GWAS results for our PNA and IDK phenotypes are available through the GWAS catalog accession nos. GCST90266936 for PNA and GCST90266935 for IDK.

## Research involving human participants, their data, or biological material

Policy information about studies with human participants or human data. See also policy information about sex, gender (identity/presentation), and sexual orientation and race, ethnicity and racism.

| | |
|---|---|
| Reporting on sex and gender | Sex was used as a covariate in all of the analyses presented in this study. |
| Reporting on race, ethnicity, or other socially relevant groupings | Genetic ancestry was determined using genetic data (using Principal Components analysis). Here we limited our sample to only individuals of European ancestry due to the statistical confounds presented by population stratification, as is standard in the literature. All GWAS and polygenic prediction exercises also controlled for genetic ancestry (the top 20 principal components of the genetic variance-covariance matrix of the genetic data for GWAS analyses and the top 10 for polygenic prediction exercises). |
| Population characteristics | Population characteristics for both samples are described in the "Behavioural & social sciences study design" section below. |
| Recruitment | The UK Biobank (UKB) is a health resource which has the purpose of improving the prevention, diagnosis, and treatment of human disease75. It consists of a prospective cohort of 502,620 men and women aged 40-69 recruited in the years 2006-2010 throughout the United Kingdom. The touchscreen questionnaire is a collection of self-reported information regarding general health, dietary habits, physical activity, psychological and cognitive states, sociodemographic factors, etc. We began with 361,194 unrelated individuals of European genetic ancestry who passed quality control measures (https://www.nealelab.is/uk-biobank/ukbround2announcement). We excluded individuals who were enrolled only in the UKB pilot study (N=335). Participants who decided to terminate the touch screen questionnaire were asked to select PNA to all subsequent questions, and they were kept in our analyses. Conversely, individuals who withdrew from the study without filling out the touchscreen survey were excluded from the analysis (N=231). As a result, a total of N=360,628 participants took part in the survey and answered every question of interest in the study; this is the final analytic sample size.<br><br>Add Health originated as an in-school survey of a nationally representative sample of US adolescents enrolled in grades 7 through 12 during the 1994-1995 school year. Respondents were born between 1974 and 1983, and a subset of the original Add Health respondents has been followed up with in-home interviews, which allows researchers to assess correlates of outcomes in the transition to early adulthood. In Add Health, the mean birth year of respondents is 1979 (SD = 1.8), and the mean age at the time of assessment (Wave 4) is 29.0 years (SD = 1.8). All phenotypes included in this study come from Wave 4, the latest wave of Add Health data collection (2007-2009). |
| Ethics oversight | Use of the UK Biobank data was approved under application 31063. Because of the sensitive nature of this study, we also explicitly sought permission for the specific scope of this paper (i.e., "to also study response rates and response characteristics (e.g., how often a response is left unanswered) and to examine whether there are any genetic factors that correlate with these response phenotypes"), which the UK Biobank granted under the same application.<br>Analysis of the UK Biobank data was reviewed by the Partners HealthCare IRB (Partners Human Research), which determined in expedited review that the project met the US federal criteria definition of "not human subjects research" (Protocol #2019P000883 titled "Behavioral Genetics Study of Responsiveness from UKBB Questionnaires").<br>Analysis of the Add Health data was reviewed by the Office of Research Subject Protection (OSRP) at the Broad Institute of MIT and Harvard, which determined that the project met US federal criteria for exemption from IRB review (Project #0001 titled "Genetic and environmental factors influencing complex social behavior"). |

Note that full information on the approval of the study protocol must also be provided in the manuscript.

# Field-specific reporting

Please select the one below that is the best fit for your research. If you are not sure, read the appropriate sections before making your selection.

☐ Life sciences  ☒ Behavioural & social sciences  ☐ Ecological, evolutionary & environmental sciences

For a reference copy of the document with all sections, see nature.com/documents/nr-reporting-summary-flat.pdf

# Behavioural & social sciences study design

All studies must disclose on these points even when the disclosure is negative.

| | |
|---|---|
| Study description | We performed the most in-depth nvestigation of item nonresponse behavior to date. We created factor scores for the two most common nonresponse answers, "Prefer Not to Answer" (PNA) AND "I Don't Know" (IDK) in our datasets. We conducted complementary phenotypic and genetic analyses to gain insights that would not otherwise have be obtainable by solely leveraging questionnaire-based information. All data was quantitative. |
| Research sample | Using the UK Biobank, we examined nonresponse behavior in 109 questionnaire items for 230,629 individuals. We performed replication exercises using polygenic scores for 3,414 individuals in the Add Health study. We chose these samples because of the large and required sample size for genetic analyses (UK Biobank) and for existence of well-phenotyped replication data (Add Health). the UK Biobank is not a nationally representative study, while the Add Health study is a US-based nationally representative study. |
| Sampling strategy | Analytic samples were decided by using the samples of subsamples that had the largest N for a given outcome under study. This strategy was chosen, because large samples are prioritized to have enough statistical power to isolate small genetic associations. This is common practice in the field. |
| Data collection | The UK Biobank (UKB) is a health resource which has the purpose of improving the prevention, diagnosis, and treatment of human disease75. It consists of a prospective cohort of 502,620 men and women aged 40-69 recruited in the years 2006-2010 throughout the United Kingdom. The touchscreen questionnaire is a collection of self-reported information regarding general health, dietary habits, physical activity, psychological and cognitive states, sociodemographic factors, etc. We began with 361,194 unrelated individuals of European genetic ancestry who passed quality control measures (https://www.nealelab.is/uk-biobank/ ukbround2announcement). We excluded individuals who were enrolled only in the UKB pilot study (N=335). Participants who decided to terminate the touch screen questionnaire were asked to select PNA to all subsequent questions, and they were kept in our analyses. Conversely, individuals who withdrew from the study without filling out the touchscreen survey were excluded from the analysis (N=231). As a result, a total of N=360,628 participants took part in the survey and answered every question of interest in the study; this is the final analytic sample size.<br><br>Add Health originated as an in-school survey of a nationally representative sample of US adolescents enrolled in grades 7 through 12 during the 1994-1995 school year. Respondents were born between 1974 and 1983, and a subset of the original Add Health respondents has been followed up with in-home interviews, which allows researchers to assess correlates of outcomes in the transition to early adulthood. In Add Health, the mean birth year of respondents is 1979 (SD = 1.8), and the mean age at the time of assessment (Wave 4) is 29.0 years (SD = 1.8). All phenotypes included in this study come from Wave 4, the latest wave of Add Health data collection (2007-2009). |
| Timing | The UK Biobank is a prospective cohort of 502,620 men and women aged 40-69 recruited in the years 2006-2010 throughout the United Kingdom.<br><br>Add Health originated as an in-school survey of a nationally representative sample of US adolescents enrolled in grades 7 through 12 during the 1994-1995 school year. Respondents were born between 1974 and 1983, and a subset of the original Add Health respondents has been followed up with in-home interviews, which allows researchers to assess correlates of outcomes in the transition to early adulthood. |
| Data exclusions | We use only individuals of European ancestry due to the statistical confounds presented by population stratification, as is standard in the literature. |
| Non-participation | Participants were able to select nonresponse options through the questionnaires in the UK Biobank and Add Health. We used these nonresponse options as the primary analysis variables in our study. |
| Randomization | Randomization was not a relevant component in this study, as there were no experimental conditions being tested and participants were divided into whether they answered questionnaires with nonresponse options or not. |

# Reporting for specific materials, systems and methods

We require information from authors about some types of materials, experimental systems and methods used in many studies. Here, indicate whether each material, system or method listed is relevant to your study. If you are not sure if a list item applies to your research, read the appropriate section before selecting a response.

## Materials & experimental systems

| n/a | Involved in the study |
|---|---|
| ☒ ☐ | Antibodies |
| ☒ ☐ | Eukaryotic cell lines |
| ☒ ☐ | Palaeontology and archaeology |
| ☒ ☐ | Animals and other organisms |
| ☒ ☐ | Clinical data |
| ☒ ☐ | Dual use research of concern |
| ☒ ☐ | Plants |

## Methods

| n/a | Involved in the study |
|---|---|
| ☒ ☐ | ChIP-seq |
| ☒ ☐ | Flow cytometry |
| ☒ ☐ | MRI-based neuroimaging |

