## [Peer Review File · Nature Human Behaviour]

Peer Review Information

Journal: Nature Human Behaviour

Manuscript Title: Patterns of item nonresponse behavior to survey questionnaires are systematic and associated with genetic loci

Corresponding author name(s): Robbee Wedow and Andrea Ganna

Reviewer Comments & Decisions:

Decision Letter, initial version:

5th March 2022

Dear Dr Wedow,

Thank you once again for your manuscript, entitled "Patterns of item nonresponse behavior to survey questionnaires are systematic and have a genetic basis," and for your patience during the peer review process.

Your manuscript has now been evaluated by 3 reviewers, whose comments are included at the end of this letter. Although the reviewers find your work to be of interest, they also raise some important concerns. We are interested in the possibility of publishing your study in Nature Human Behaviour, but would like to consider your response to these concerns in the form of a revised manuscript before we make a decision on publication.

To guide the scope of the revisions, the editors discuss the referee reports in detail within the team, including with the chief editor, with a view to (1) identifying key priorities that should be addressed in revision and (2) overruling referee requests that are deemed beyond the scope of the current study. We hope that you will find the prioritised set of referee points to be useful when revising your study. Please do not hesitate to get in touch if you would like to discuss these issues further.

- 1) We agree with our reviewers that your manuscript should demonstrate the practical utility of your findings. In doing so, however, please ensure that ethical considerations are taken into account.
- 2) Please ensure that your analytical approach is technically appropriate, particularly with regard to the first point made by Reviewer 2 about the appropriateness of creating a single score for non-response rather than domain-specific scores.
- 3) Please engage more fully with existing literature.

4) Please provide full ethics information at the start of your Methods section. This should include full information on the protocol under which approval was provided by UK Biobank and Add Health to carry out these analyses, as well as ethics approval for the project itself.

In sum, we invite you to revise your manuscript taking into account all reviewer and editor comments. We are committed to providing a fair and constructive peer-review process. Do not hesitate to contact us if there are specific requests from the reviewers that you believe are technically impossible or unlikely to yield a meaningful outcome.

We hope to receive your revised manuscript within two months. We understand that the COVID-19 pandemic is causing significant disruption for many of our authors and reviewers. If you cannot send your revised manuscript within this time, please let us know - we will be happy to extend the submission date to enable you to complete your work on the revision.

- Include a "Response to the editors and reviewers" document detailing, point-by-point, how you addressed each editor and referee comment. If no action was taken to address a point, you must provide a compelling argument. This response will be used by the editors to evaluate your revision and sent back to the reviewers along with the revised manuscript.
- Highlight all changes made to your manuscript or provide us with a version that tracks changes.

[REDACTED]

We look forward to seeing the revised manuscript and thank you for the opportunity to review your work. Please do not hesitate to contact me if you have any questions or would like to discuss these revisions further.

Sincerely,

Charlotte Payne

Charlotte Payne, PhD
Senior Editor

Nature Human Behaviour

Reviewer expertise:

Reviewer #1: GWAS; bias in GWAS

Reviewer #2: Bias in biobanks; epidemiology

Reviewer #3: GWAS, PGSs

REVIEWER COMMENTS:

Reviewer #1:

Remarks to the Author:

Mignogna et al. presented an important and interesting study on the genetic basis of nonresponse to survey questionnaires. The factor scores of two main traits were analysed: "Prefer not to answer" (PNA) and "I don't know" (IDK), composed of 109 questionnaire items from the UK Biobank (UKB). They found that PNA and IDK were highly genetically correlated with one another and the nonresponse were correlated with lower educational attainment and poorer health status. The studies also identified the specific or shared genetic loci associated with PNA and IDK.

This is a timely report that provides a systematic perspective of the genetic basis of nonresponse. The manuscript is clearly written and the evidence are adequate to support the conclusion. I have several comments and hope they are helpful to improve the manuscript.

Major comments:

The first concern of the study design on the definition of nonresponse. While PNA and IDK were coded as -3 and -1 in the data, can the participants in UKB somehow skip (maybe by directly pressing Next button) any questions? If so, how is this recorded in the phenotype file, by "NAs"?

In page #13, the authors claimed "Questions asked to a subset of participants conditional on their answer to other questions were excluded." If we take a close look at Page #22 of the questionnaire (<https://biobank.ndph.ox.ac.uk/showcase/ukb/docs/TouchscreenQuestionsMainFinal.pdf>), the smoking status (ID=20116) is actually a summary of current tobacco smoking (ID=1239) and past tobacco smoking (ID=1249), and the past tobacco smoking is a conditional question on the current tobacco smoking. In this case, the smoking status seems not to fit the definition because statistically speaking it is a perfect linear combination of the two questions. This is consistent with the Supe Table 2 that "number of individuals choosing PNA" for smoking status is 1,307 and for currently smoking is 218, which implies for past smoking is 1,089.

What is the hypothesis that PNA is less common than IDK? In the UKB touchscreen survey, it seems the

option IDK is always put ahead of PNA (please correct me if I'm wrong). Is that possible many participants prefer not to answer but when they saw the IDK option, they feel more comfortable to select IDK? The social desirability may not only affect the choice of response or nonresponse, but could also unintentionally drive the participants to select IDK rather than PNA. Could that also be the reason why these traits are strongly genetically correlated and why the 3 of 4 GWAS loci of PNA also significant in IDK?

I am glad that the authors include the "the online follow-up dietary questionnaires" and predicted the participation of this follow-up. However, in the whole online follow-up questionnaire (Category 100089), there are many more sensitive questions like addiction, alcohol/cannabis use, and self-harm behaviors. Have the authors also tried to investigate the non-response pattern for these traits? Given online follow-up is also an ascertained group, comparing the PNA/IDK percentage between the initial assessment and online follow-up on the similar question will be even helpful to understand the nonresponse pattern.

The childhood sunburn example is quite interesting. The allele that is associated with a higher risk of not knowing how many times someone was sunburned as a child, also increases the risk for melanoma and cutaneous squamous cell carcinoma. If "not knowing" is due to "not remember", the loci might be related to cognitive function. However, not knowing the sunburn is positively related to melanoma risk. Does this result suggests this small group of participants was intentionally or unintentionally conceal their sunburnt experience as a child? Previous study has investigated individual item-level values of non-responders in alcohol drinking/tobacco smoking/physical activity (<https://doi.org/10.1038/s41467-020-20237-6>) and found the nonresponse pattern related to both disease outcome and social-economic status. Could the authors provide some insights into this discrepancy?

Is social deprivation also included in this study? In Table 1, the demographics about PNA and IDK non-responders are obviously region-dependent. Social deprivation is measured by Townsend deprivation index in the UKB (field ID=189), and this index is corresponding to the output area in which their postcode is located. It should be considered as a major element of SES along with educational attainment and household income. Please also note that this index captures the variations in employment status.

Minor comments:

While only 83 out of 109 questions allowed the IDK option, are those remaining 26 questions enriched in any categories? I wonder why the design of those questions don't have IDK options.

In page #9 where authors mentioned " IDK-adjusted PNA became associated with bipolar disorder and schizophrenia ", does that mean the estimates are not significant before adjustment? The estimates should also be included here for comparison.

The authors may be aware of a recent study on the genetics of participation (<https://doi.org/10.1101/2022.02.11.480067>). Could authors spend one or two words on how their results are connected this study? Are the correlation of participation and nonresponse with other target traits largely consistent in the direction?

The "psuedo" on page #6 was misspelt.

In Figure 3, annotating gene names of the top SNPs in the Miami plot would be helpful.

If we combine the Figure 4 & 5 and contrast of rg estimates between IDK and PNA for the same traits, what traits will show large difference?

Sorry if I missed anything. The authors mentioned stratified LD Score regression but I didn't see any results from this analysis.

Reviewer #2:

Remarks to the Author:

Key results: The authors proposed to “clarify the contribution of genetics to differences in item nonresponse behaviors between individuals.” To this end, they constructed latent factor scores for reporting “prefer not to answer” (PNA) and “I don’t know” (IDK) to 109 questions from the UK Biobank. Next, they conducted GWAS (sex, age, 20 PC-adjusted) on these scores, for which they found 4 and 35 genome-wide significant hits for PNA and IDK, respectively. Identified SNPs have previously been associated with education, intelligence, mental disorders, and gastrointestinal disease (Crohn’s/IBS). Heritability estimates (2-7%) were significant, but an order of magnitude less than previous reports (18-32% in Taylor, et al. 2018).

Through several approaches (LD Score Regression, Genomic SEM), authors show some genetic associations between IDK and PNA adjusted for other factors such as education, self-reported health, and the respective other score (IDK or PNA), and note correlations differ between the two non-response types. Finally, authors use IDK- and PNA-associated SNPs to construct respective polygenic scores in the Add Health longitudinal cohort / panel study and associate these scores to actual non-response in the study.

Validity: This study takes an interesting approach to constructing common, latent scores for item non-response, correlates them to SNPs, and in turn relates these signals to other traits and phenotypes. In and of themselves, these findings are interesting and takes a slightly different angle to missing data in UK Biobank than other recent papers. That said, based on the existing literature on this topic, the interpretation and utility of these genetic correlations are unclear. Moreover, there was no treatment of how these findings could improve assessment and correction for nonresponse biases as promised in the introduction. I discuss further below.

Originality and significance:

The authors begin by proposing that non-responses represent multiple constructs including comprehension/cognitive traits and/or affect/socioemotional traits. Further, they assert that in understanding determinants of non-response, these can inform generalizability of findings, and specifically that knowledge of genetic correlations can assist in modelling missingness mechanisms. These seem like sound premises and certainly address topic of great concern in the use of large, “big data” biorepositories to describe, predict, and infer on disease etiology.

The inferential consequences of voluntary participation/opt-in to cohorts and biorepositories in general (e.g. Munafò, et al. 2018; Richiardi, et al. 2019; Beesley, Salvatore, et al. 2020) and the UK Biobank specifically (Swanson, 2012; Fry, et al. 2017; Keyes and Westreich 2019; Haworth, et al. 2019; Batty, et al. 2020; Stamatakis, et al. 2021; Huang, 2021) have been extensively discussed. Recently, the relevance of potential selection biases due to nonprobability sampling on genetic associations has been highlighted, again in general (Beesley, Fritsche, et al. 2020; Taylor, et al. 2018) and specifically with respect to the UK Biobank (Haworth, et al. 2019; Sohail, et al. 2019; Huang, 2021). The major upshot of this body of work is that genetic correlations themselves may be biased if both a factor related to genotype (e.g. geographic region) and a phenotype of interest (e.g. mental health) lead to differential participation, in a sense introducing spurious population stratification. While past works have shown how phenotypes influence response, the most recent works show that differential participation lead to extensive spurious genotype correlations, e.g. Haworth, et al. showed strong geographic associations even after correction for 40 PCs. In other words, the correlations the authors use to demonstrate potential causes for nonresponse may themselves be biased due to voluntary participation in the UK Biobank (and the genetic component) itself, and cannot be teased out from true genetic "causes" of non-response.

That said, in looking at different item response, this study does indeed have the potential to go further than previous studies in investigating the potential sources of both non-response, potential behavioral correlates, and the impact of non-response biases. In particular, as inferential targets may be biased to different degrees depending on the specific genotypes, phenotypes, and selection/non-response processes under study, the authors are correct that understanding the different mechanisms of non-response would be worthwhile. However, this would require the authors to engage with the possibility that genetic correlations are subject to selection biases. Importantly, the authors would also have to present some strategy to use these findings to correct for non-response bias, which is missing from the manuscript. Importantly, distinguishing when correlations are truly spurious or when they might represent a mediating mechanism (i.e. genotype \rightarrow education \rightarrow non-response) is needed. Diagramming the targets of inference and how selection bias may occur (Huang 2021), propensity score weighting and standardization (Stamatakis, et al. 2021), likelihood-based correction techniques (Beesley, et al), or simply quantifying sensitivity to non-response bias (Smith and VanderWeele 2019) are all potential components of this solution.

With respect to the goal of better characterizing non-response, there were two other analytic choices that appeared to be questionable:

1. Given non-response is suspected to be a product of several constructs, and therefore multiple sources of missingness, what is the motivation behind only creating one score (accounting for substructure) for each non-response? Even leaving aside individual-item GWAS for privacy purposes, would it not make sense to estimate associations with each a priori or exploratory domain/class of questions, separately? Wouldn't this be the most effective way to address the specific selection/missingness mechanisms for each downstream hypothesis? This would also preempt any questions about your modelling decisions / pipeline for your full and reduced bi-factor FA or the relevance of factors that only explain 51% and 26% of the overall non-response. SNPs poorly predicted an overall summary score, but perhaps heritability estimates would be higher if scores were aggregated in other domain/construct specific ways?

A related point: The authors state that they intend to avoid analyses that violate participants desire for non-response, which seems an admirable and ethical goal. However, how does this square with the

potential use of genetic scoring to model missingness mechanisms, which in effect, attempt to restore information about the participants response (or at least response type)? The author allude to this challenge when they say that one issue of non-response is a reduction of the effective sample size. Presumably the restoration of effective sample size comes from imputation of plausible values for individuals with similar genetic scores on these items, or somewhat equivalently, upweighting of individuals with non-missing response but otherwise similar characteristics (including genotype)?

It seems with respect to ethical considerations, in balance, it would be a greater imperative to correct the biases arising from misattributing genetic correlations, wherein say intergenerational disadvantage leads to differential participation in the genetic component and item response and thus reintroduction of population stratification due to selection/collider bias. These biases may in turn re-inforce/re-if perceptions around the relative contribution of genetics to education, mental capacity, "compliance" or reticence to participate in research, and other inferences that may arise from less-than-careful interpretations of these genetic correlations.

2. Further, with respect to your study objectives, why is it important that the factor score for non-response increases prediction of future participation over other "risk factors" (0.3%, at that)? Is it not sufficient that genotype associate with nonresponse including education and health as genetic "causes" of non-response may be mediated through these? I suspect that further elaboration and specification of a proposed solution to non-response would then inform the utility of these residual correlations.

REFERENCES

Munafò MR, Tilling K, Taylor AE, Evans DM, Davey Smith G. Collider scope: when selection bias can substantially influence observed associations. *Int J Epidemiol*. 2018;47:226–235.

Griffith GJ, Morris TT, Tudball MJ, et al. Collider bias undermines our understanding of COVID-19 disease risk and severity. *Nat Commun*. 2020;11:5749.

Haworth S, Mitchell R, Corbin L, et al. Apparent latent structure within the UK Biobank sample has implications for epidemiological analysis. *Nat Commun*. 2019;10:333.

Huang JY. Representativeness Is Not Representative: Addressing Major Inferential Threats in the UK Biobank and Other Big Data Repositories. *Epidemiology*. 2021 Mar 1;32(2):189-193.

Keyes KM, Westreich D. UK Biobank, big data, and the consequences of non-representativeness. *Lancet*. 2019;393:1297.

Richiardi L, Pearce N, Pagano E, Di Cunzio D, Zugna D, Pizzi C. Baseline selection on a collider: a ubiquitous mechanism occurring in both representative and selected cohort studies. *J Epidemiol Community Health*. 2019;73:475–480.

Smith LH, VanderWeele TJ. Bounding bias due to selection. *Epidemiology*. 2019;30:509–516.

Sohail M, Maier RM, Ganna A, et al. Polygenic adaptation on height is overestimated due to uncorrected stratification in genome-wide association studies. *Elife*. 2019;8:e39702.

Stamatakis E, Owen KB, Shepherd L, Drayton B, Hamer M, Bauman AE. Is cohort representativeness passé? Matching the UK Biobank sample to source population characteristics and recalculating the associations between lifestyle risk factors and mortality. *Epidemiology*. 2021 Mar 1;32(2):179-188.

Swanson JM. The UK Biobank and selection bias. *Lancet*. 2012;380:110.

Taylor AE, Jones HJ, Sallis H, et al. Exploring the association of genetic factors with participation in the Avon Longitudinal Study of Parents and Children. *Int J Epidemiol*. 2018;47:1207–1216.

Data & methodology:

The study is generally well described, particularly the construction of the latent non-response factors through EFA / CFA. As alluded to above, engagement with the past literature on selection bias in UK Biobank and implications on genotypic associations is needed. These associations need not be causal: Selection bias into contribution of genetic material can also explain these findings, even if restricted to a single self-reported ancestry (European).

Additionally, more details on the LD Score Regressions and PRS constructions would be appreciated both in the main methods and as supplements. E.g. which / how many SNPs were chosen, what were the thresholds for selection, etc.

Preregistration: N/A

Appropriate use of statistics and treatment of uncertainties: See above.

Custom code: Scripts provided; not evaluated (no source data)

Conclusions: Clearer elaboration of the use of these genetic correlations to understanding and correcting non-response bias is needed. This will entail engaging with the selection bias literature a bit more closely. Relatedly, the authors have the potential to improve understanding of non-response bias over existing literature by making more use of domains/patterns in non-response, rather than opting for a single underlying construct.

Suggested improvements: Described above.

References: Additional suggestions listed above.

Clarity and context: The abstract and manuscript are clearly written.

- Jonathan Huang (jonathan_huang@sics.a-star.edu.sg)

Reviewer #3:

Remarks to the Author:

Thanks for the opportunity to read the manuscript "Patterns of item nonresponse behavior to survey questionnaires are systematic and have a genetic basis". As the title suggests, this is a study of genetically linked nonresponse patterns in order to reduce issues with MNAR in respect to latent variables.

I believe this is a very interesting contribution, well designed, conducted, and written, and it merits potentially a high-impact publication, but I would challenge the authors for two main points and two minor ones.

Main points:

1. First, item non-response is a very, very broad and well-established field of research and I believe that this study could build more on this literature on all levels. First, the main mediators of genetic effects on non-response appear to be education and health and there should be a theory about why this is the case. Second, there must be further literature (I'm not an expert and would recommend one as a reviewer) on the main predictors of item non-response which needs review, and the findings need to be related to existing knowledge. Finally, the contributions of this study which I see so far are first that it can measure a "latent" "disposition" for non-response in contrast to observed predictors. This contribution could be further quantified beyond the revisualization of education and health. It would also be interesting to see the summary statistics revisualized for the genetic correlations with other traits which are potentially measurable and the resulting heritability of the non-response GWAS. While the theoretical contribution of receiving a yet unmeasured predictor for item-nonresponse is clear and great, it should be made very clear as well what the practical/empirical contribution is, how much variance we explain on top of measurable variables or genetically explored ones.

2. I am really wondering about the practical value of the knowledge of this selection variable. My intuition is that it can be used for some sort of conditional/selection regression analysis a la Heckman. Can't this be demonstrated, for example, in the Add Health prediction sample, e.g. including the selection score in a Heckman model of the education score predicting educational attainment? What else will be the use?

Minor

3. Another very interesting feature is the high genetic correlation between IDK and PNA, as if this is partly the same trait? I was really wondering what that means – could you interpret it? Furthermore, there is a conditional genetic correlation analysis of both items with other traits which is interesting, but would it also be possible to receive some sort of ranking of these categories in a genetic correlation analysis with, for example, education? As in: IDK scores higher for education than PNA.

4. Maybe it would be good to explicitly mention that the binary predictions are based on linear probability models.

5. The manuscript needs careful editing.

Author Rebuttal to Initial comments

Reviewer #1

Mignogna et al. presented an important and interesting study on the genetic basis of nonresponse to survey questionnaires. The factor scores of two main traits were analysed: “Prefer not to answer” (PNA) and “I don’t know” (IDK), composed of 109 questionnaire items from the UK Biobank (UKB). They found that PNA and IDK were highly genetically correlated with one another and the nonresponse were correlated with lower educational attainment and poorer health status. The studies also identified the specific or shared genetic loci associated with PNA and IDK.

This is a timely report that provides a systematic perspective of the genetic basis of nonresponse. The manuscript is clearly written and the evidence are adequate to support the conclusion. I have several comments and hope they are helpful to improve the manuscript.

Response:

We very much thank the reviewer for these thoughtful and kind remarks. We have done our best to incorporate all of the proposed changes, which we highlight below.

Comment 1:

The first concern of the study design on the definition of nonresponse. While PNA and IDK were coded as -3 and -1 in the data, can the participants in UKB somehow skip (maybe by directly pressing Next button) any questions? If so, how is this recorded in the phenotype file, by "NAs"?

Response:

We thank the reviewer for pointing out that we needed to clarify this definition. The UKB touchscreen protocol (<https://biobank.ndph.ox.ac.uk/showcase/ukb/docs/TouchscreenQuestionsMainFinal.pdf>; <https://biobank.ndph.ox.ac.uk/showcase/ukb/docs/Touchscreen.pdf>) strongly implies, though doesn't directly state, that this kind of skipping without providing a coded response would have been prevented by the logical validations implemented in their ACE system. For our purposes we only used items that had valid responses (whether substantive or IDK/PNA), so any items that allowed NAs from skipping or other missingness that didn't involve an affirmative nonresponse answer are excluded from this analysis. This means we exclude, for example, (a) items asked only to a subset of participants; (b) groups of questions like the sexual history questionnaire in UKB that allow the participant to skip the entire section; or

(c) items added to the touchscreen questionnaire later in the recruitment process. We clarified these inclusion/exclusion specifics in the main text and in the “PNA/IDK definitions” section of the Methods.

Comment 2:

In page #13, the authors claimed "Questions asked to a subset of participants conditional on their answer to other questions were excluded." If we take a close look at Page #22 of the questionnaire

(<https://biobank.ndph.ox.ac.uk/showcase/ukb/docs/TouchscreenQuestionsMainFinal.pdf>), the smoking status (ID=20116) is actually a summary of current tobacco smoking (ID=1239) and past tobacco smoking (ID=1249), and the past tobacco smoking is a conditional question on the current tobacco smoking. In this case, the smoking status seems not to fit the definition because statistically speaking it is a perfect linear combination of the two questions. This is consistent with the Supe Table 2 that "number of individuals choosing PNA" for smoking status is 1,307 and for currently smoking is 218, which implies for past smoking is 1,089.

Response:

We thank the reviewer for identifying this important edge case. Indeed, Smoking status (20116) is a summary of both current (1239) and past (1249) smoking, and past smoking is conditional on current. In other words, a participant only gets asked about past smoking if that participant is *not* a current tobacco smoker.

We have revised our definition on Page 13 to read that “These Items were included in our analyses only if valid responses existed for all participants in our sample,” which is why this derived item was indeed included, and added a note that our definition “includes one instance of a derived item that retains nonresponse information (UKB FieldID 20116)”. We have also checked and confirmed the remainder of our included items to ensure that this is indeed the only edge case like this one.

Comment 3:

What is the hypothesis that PNA is less common than IDK? In the UKB touchscreen survey, it seems the option IDK is always put ahead of PNA (please correct me if I'm wrong). Is that possible many participants prefer not to answer but when they saw the IDK option, they feel more comfortable to select IDK? The social desirability may not only affect the choice of response or nonresponse but could also unintentionally drive the participants to select IDK rather than PNA. Could that also be the reason why these

traits are strongly genetically correlated and why the 3 of 4 GWAS loci of PNA also significant in IDK?

Response:

The reviewer is correct that IDK is always put ahead of PNA, and we agree social desirability could play some role in the correlation of our traits. Although the UKB data only provides a forced choice between IDK and PNA and we can't fully disentangle what is happening in participants' minds when they choose IDK vs.

PNA, the analyses of difference in genetic correlations with outside traits conditional on the other nonresponse option provide some insight into the differential use of the nonresponse options. We've updated the main text to provide more discussion of these points about the IDK/PNA rates and the observed overlap, and added additional analysis trying to clarify where PNA and IDK genetic effects differ, which highlight the psychological/cognitive aspect of these nonresponse behaviors.

Comment 4:

I am glad that the authors include the "the online follow-up dietary questionnaires" and predicted the participation of this follow-up. However, in the whole online follow-up questionnaire (Category 100089), there are many more sensitive questions like addiction, alcohol/cannabis use, and self-harm behaviors. Have the authors also tried to investigate the non-response pattern for these traits? Given online follow-up is also an ascertained group, comparing the PNA/IDK percentage between the initial assessment and online follow-up on the similar question will be even helpful to understand the nonresponse pattern.

Response:

We appreciate the suggestion to compare PNA/IDK rates in the follow-up vs. the baseline questionnaire. In the mental health follow-up we observe 25.3% with at least one IDK response and 10.1% with at least one PNA, compared to 67% and 8.8% at baseline, respectively. We've added this result to the paper, noting the relevance of additional ascertainment

Although we also agree that many potentially interesting questions could be asked about the nonresponse pattern for those items (and many others), we have chosen for this paper to avoid investigating anything too domain-specific or item-specific for ethical reasons, in an attempt to protect the voluntary decision of participants to not answer

certain specific questions. To address the question of nonresponse to sensitive items we instead focus on association with two summaries of nonresponse across sensitive items: nonresponse to 1+ items in the mental health, and choosing to skip the full sexual history questionnaire.

Comment 5:

The childhood sunburn example is quite interesting. The allele that is associated with a higher risk of not knowing how many times someone was sunburned as a child, also increases the risk for melanoma and cutaneous squamous cell carcinoma. If "not knowing" is due to "not remember", the loci might be related to cognitive function. However, not knowing the sunburn is positively related to melanoma risk. Does this result suggest this small group of participants was intentionally or unintentionally conceal their sunburned experience as a child? Previous study has investigated individual item-level values of non-responders in alcohol drinking/tobacco smoking/physical activity (<https://doi.org/10.1038/s41467-020-20237-6>) and found the nonresponse pattern related to both disease outcome and social-economic status. Could the authors provide some insights into this discrepancy?

Response:

We agree the sunburn result is quite interesting, and the reviewer pose some great questions. We realize that our original draft only described our interpretation of the technical check without elaborating on our hypothesized understanding of the substantive results; we've updated the manuscript to add this interpretation.

Specifically, while "not knowing" will in some cases reflect cognitive function as the reviewer describes, we anticipate that there will be a substantial trend that the more times a person was sunburned the harder it will be for the person to recall the number of sunburn occasions. In other words, if someone never been sunburned then it's easy to answer 0 (and similarly for 1-2 sunburns being easy to remember as notable), but if they've been sunburned dozens of times then they may struggle to recall the exact number and choose to report IDK rather than some uncertain estimate. As a result individuals who are more (genetically) predisposed to sunburn - and thus eventual melanoma - will be more likely to respond IDK.

This hypothesis is generally consistent with the Xue et al. findings for alcohol/tobacco/etc, in that it suggests individuals in the disease state (e.g. melanoma or alcohol dependence) will have both higher and more variable levels of the risk variable (sunburns or alcohol consumption) and as a result higher likelihood of

misreporting/nonresponse. We do however expect that sunburn carries much weaker social desirability concerns than substance use, and thus the IDK responses for sunburn are much less likely to be motivated by intentional concealment.

Comment 6:

Is social deprivation also included in this study? In Table 1, the demographics about PNA and IDK non-responders are obviously region-dependent. Social deprivation is measured by Townsend deprivation index in the UKB (field ID=189), and this index is corresponding to the output area in which their postcode is located. It should be considered as a major element of SES along with educational attainment and household income. Please also note that this index captures the variations in employment status.

Response:

This is an important observation. Accordingly, we have re-run our GenomicSEM to include the Townsend deprivation index as an additional control; these results are now highlighted in the “Genetic correlations with heritable traits” along with the “GenomicSEM” section of Methods. We also highlight these results in Supplementary Figure 8 and Supplementary Table 9, where we still find that “These results suggest that even after accounting for the genetic associations with socioeconomic factors, genetic associates with nonresponse were shared with genetic associations for poor overall physical and mental health.” The results are not substantively different, but we appreciate the opportunity to add this additional control to improve the robustness of the interpretation.

Comment:

While only 83 out of 109 questions allowed the IDK option, are those remaining 26 questions enriched in any categories? I wonder why the design of those questions don't have IDK options.

Response:

We are as interested as the reviewer is in this question. While we cannot be certain, it appears most of the 26 questions that did not allow the IDK option are in areas where it might be deemed unlikely a participant wouldn't know the answer:

- Health problems, (e.g. mouth/teeth dental problems, eye problems, pain experiences in last month, vascular/heart problems diagnosed by doctor, blood clot, bronchitis, asthma, etc.)
- Socioeconomic questions (e.g. qualifications, current employment status, type

- of accommodations lived in)
- Lifestyle (e.g. frequency of social activities or playing video games)

The PNA option would still be included in those cases to allow participants to decline to answer if they would prefer not to.

Comment:

In page #9 where authors mentioned " IDK-adjusted PNA became associated with bipolar disorder and schizophrenia ", does that mean the estimates are not significant before adjustment? The estimates should also be included here for comparison.

Response:

It's correct that IDK is not significantly genetically correlated with schizophrenia ($r_g = -.0055$, $p = 0.81$) or bipolar ($r_g = -.0922$, $p = 7.8e-5$) prior to adjustment for PNA. This occurs because PNA shows a nominal positive correlation with schizophrenia/bipolar, enough so that the limited negative correlation observed for IDK implies a crossover effect where the residual component of IDK not explained by PNA has a stronger negative correlation.

The updated draft tries to make these comparisons clearer by following the reviewer's later suggestion (also recommended by Reviewer #3) to focus the analysis on describing where IDK and PNA have differing correlations with a trait. For schizophrenia and bipolar, we report that their respective correlations with IDK differ from their correlations with PNA, and that conditional analysis (i.e., adjustment of IDK for PNA and vice versa as reported as the primary result previously) suggests a crossover effect with unique genetic components of PNA positively correlated with schizophrenia/bipolar while the unique genetic components of IDK are negatively correlated.

Comment:

The authors may be aware of a recent study on the genetics of participation (<https://doi.org/10.1101/2022.02.11.480067>). Could authors spend one or two words on how their results are connected this study? Are the correlation of participation and nonresponse with other target traits largely consistent in the direction?

Response:

We now explicitly discuss this study, which became available shortly after we submitted our manuscript, in the discussion of our main text.

Comment:

The "psuedo" on page #6 was misspelt.

Response:

This has been corrected.

Comment:

In Figure 3, annotating gene names of the top SNPs in the Miami plot would be helpful.

Response:

This has now been added to the Miami plot. We have also now added these gene names to Supplementary Tables 5 and 6.

Comment:

If we combine the Figure 4 & 5 and contrast of r_g estimates between IDK and PNA for the same traits, what traits will show large difference?

Response:

We appreciate this excellent suggestion. We have added new analyses to compare r_g estimates as described, and find them quite fruitful for clarifying the comparison. We have combined Figures 4 and 5 into a single figure with Panels a and b, and have also added panel c to illustrate these results, which are also now included in Supplementary Table 8. In addition, we now note in the main text:

*"To help understand which genetic components are unique to PNA and IDK we considered traits whose correlation with the two nonresponse types differed (**Suppl. Fig. 8c**). Of the 654 additional traits tested, 38 had significantly different genetic correlation estimates for PNA and IDK (**Suppl. Tab. 8; Fig 5c**). Among these 38 phenotypes, PNA had stronger genetic correlations with psychiatric (e.g., schizophrenia $r_{g_PNA}=0.21(0.03)$ vs. $r_{g_IDK}=-0.006(0.02)$, $p_{diff}=3.72 \times 10^{-12}$), cognitive (e.g., general cognitive performance $r_{g_PNA}=-0.46(0.03)$ vs. $r_{g_IDK}=-0.27(0.02)$, $p_{diff}=3.33 \times 10^{-12}$), and sociodemographic variables (e.g., educational attainment $r_{g_PNA}=-0.51(0.03)$ vs. $r_{g_IDK}=-0.38(0.02)$,*

$p_{diff}=2.05 \times 10^{-8}$), while IDK showed more substantial correlation with reported activities (e.g. using UV protection $r_{g_IDK}=-0.12(0.03)$ vs. $r_{g_PNA}=0.05(0.02)$, $p_{diff}=1.41 \times 10^{-6}$) and nutrition (salad intake $r_{g_IDK}=-0.21(0.03)$ vs. $r_{g_PNA}=-0.02(0.04)$, $p_{diff}=5.83 \times 10^{-7}$).

Comment:

Sorry if I missed anything. The authors mentioned stratified LD Score regression but I didn't see any results from this analysis.

Response:

We have made this more clear, highlighting these results in the main text in the "Genome-wide association study (GWAS) of item nonresponse" section and in Supplementary Table 7. We have also discussed our use of LD Score Regression more specifically in the Methods in the "Heritability and enrichment" section.

Reviewer #2**Comment:**

Key results: The authors proposed to "clarify the contribution of genetics to differences in item nonresponse behaviors between individuals." To this end, they constructed latent factor scores for reporting "prefer not to answer" (PNA) and "I don't know" (IDK) to 109 questions from the UK Biobank. Next, they conducted GWAS (sex, age, 20 PC-adjusted) on these scores, for which they found 4 and 35 genome-wide significant hits for PNA and IDK, respectively. Identified SNPs have previously been associated with education, intelligence, mental disorders, and gastrointestinal disease (Crohn's/IBS). Heritability estimates (2-7%) were significant, but an order of magnitude less than previous reports (18-32% in Taylor, et al. 2018).

Through several approaches (LD Score Regression, Genomic SEM), authors show some genetic associations between IDK and PNA adjusted for other factors such as education, self-reported health, and the respective other score (IDK or PNA), and note correlations differ between the two non-response types. Finally, authors use IDK- and PNA-associated SNPs to construct respective polygenic scores in the Add Health longitudinal cohort / panel study and associate these scores to actual non-response in the study.

Response:

We thank the reviewer for this excellent and concrete summary of our work before the revision.

Comment:

Validity: This study takes an interesting approach to constructing common, latent scores for item non-response, correlates them to SNPs, and in turn relates these signals to other traits and phenotypes. In and of themselves, these findings are interesting and takes a slightly different angle to missing data in UK Biobank than other recent papers. That said, based on the existing literature on this topic, the interpretation and utility of these genetic correlations are unclear. Moreover, there was no treatment of how these findings could improve assessment and correction for nonresponse biases as promised in the introduction. I discuss further below.

Response:

We thank the reviewer for pointing out the key differences between our study and other studies in terms of our focus. We have taken the opportunity in this revision to both (a) more thoroughly discuss the theoretical interpretation of our results and how they build upon the existing literature, and (b) provide an initial empirical demonstration of the potential impact of using our results to adjust for missingness from nonresponse in GWAS. Specifically, on the former point we have extended the introduction and discussion to more clearly put our work in the context of the literature to date about participation in research cohorts, biobanks, and the UKB in particular, including diagramming the directed graph of potentially influences on item-level missingness as recommended by the reviewer's 2021 paper. On the latter point, we add new results demonstrating the use of Heckman correction as proof-of-concept for using our results to "correct" GWAS for nonresponse bias (as suggested by Reviewer 3).

Comment:

Originality and significance:

The authors begin by proposing that non-responses represent multiple constructs including comprehension/cognitive traits and/or affect/socioemotional traits. Further, they assert that in understanding determinants of non-response, these can inform generalizability of findings, and specifically that knowledge of genetic correlations can assist in modelling missingness mechanisms. These seem like sound premises and certainly address topic of great concern in the use of large, "big data" biorepositories to describe, predict, and infer on disease etiology.

The inferential consequences of voluntary participation/opt-in to cohorts and biorepositories in general (e.g. Munafò, et al. 2018; Richiardi, et al. 2019; Beesley, Salvatore, et al. 2020) and the UK Biobank specifically (Swanson, 2012; Fry, et al. 2017; Keyes and Westreich 2019; Haworth, et al. 2019; Batty, et al. 2020; Stamatakis, et al. 2021; Huang, 2021) have been extensively discussed. Recently, the relevance of potential selection biases due to nonprobability sampling on genetic associations has been highlighted, again in general (Beesley, Fritsche, et al. 2020; Taylor, et al. 2018) and specifically with respect to the UK Biobank (Haworth, et al 2019; Sohail, et al. 2019; Huang, 2021). The major upshot of this body of work is that genetic correlations themselves may be biased if both a factor related to genotype (e.g. geographic region) and a phenotype of interest (e.g. mental health) lead to differential participation, in a sense introducing spurious population stratification. While past works have shown how phenotypes influence response, the most recent works show that differential participation lead to extensive spurious genotype correlations, e.g. Haworth, et al. showed strong geographic associations even after correction for 40 PCs. In other words, the correlations the authors use to demonstrate potential causes for nonresponse may themselves be biased due to voluntary participation in the UK Biobank (and the genetic component) itself, and cannot be teased out from true genetic "causes" of non-response.

That said, in looking at different item response, this study does indeed have the potential to go further than previous studies in investigating the potential sources of both non-response, potential behavioral correlates, and the impact of non-response biases. In particular, as inferential targets may be biased to different degrees depending on the specific genotypes, phenotypes, and selection/non-response processes under study, the authors are correct that understanding the different mechanisms of non-response would be worthwhile. However, this would require the authors to engage with the possibility that genetic correlations are subject to selection biases. Importantly, the authors would also have to present some strategy to use these findings to correct for non-response bias, which is missing from the manuscript. Importantly, distinguishing when correlations are truly spurious or when they might represent a mediating mechanism (i.e. genotype -> education -> non-response) is needed. Diagramming the targets of inference and how selection bias may occur (Huang 2021), propensity score weighting and standardization (Stamatakis, et al. 2021), likelihood-based correction techniques (Beesley, et al), or simply quantifying sensitivity to non-response bias (Smith and VanderWeele 2019) are all potential components of this solution.

Response:

This was an incredibly helpful set of comments that helped us think clearly through the unique aspects of our approach that separate our work from previous work. First, we are

very thankful for the reviewer's encouraging further engagement the extensive body of literature. We have thoroughly read each cited piece, and have enthusiastically taken the opportunity to discuss this literature in our revised introduction.

We also use this literature to showcase how our approach, as explained by the reviewer is unique "as inferential targets may be biased to different degrees depending on the specific genotypes, phenotypes, and selection/non-response processes under study." We specifically adopt the reviewers suggestion of diagramming the targets of inference in the case of genetic correlation and how selection bias may occur, using our factor approach in the DAG directly. This is now Figure 1 in our paper, and we discuss the model in the introduction and in relation to our factor model results.

Next, we take the reviewer's advice (along with the advice of the editors and Reviewer 3) and attempt to demonstrate the utility of our results for correcting item-level nonresponse in GWAS studies. After reviewing all of the possible suggestions proposed by the reviewer and Reviewer 3, we evaluated a proof-of- concept Heckman correction to our GWAS of PNA and IDK. In this added analysis we observe significant changes in GWAS results and resulting genetic correlations as a result of correcting for item-level nonresponse in both constructs. In supplemental material, we also demonstrate the impact of nonresponse in a phenotypic Heckman correction. These materials are discussed in the main text and in Figure 1, in methods, in the supplement, and are also highlighted Supplementary Tables 12, 13, 14. The discussion highlights the path forward for using this kind of correction as well as the other methods the reviewer suggested for potentially understanding/quantifying the potential impacts of selection bias. We genuinely thank the reviewer for these insightful comments, which have greatly strengthened our paper.

Comment:

With respect to the goal of better characterizing non-response, there were two other analytic choices that appeared to be questionable:

1. Given non-response is suspected to be a product of several constructs, and therefore multiple sources of missingness, what is the motivation behind only creating one score (accounting for substructure) for each non-response? Even leaving aside individual-item GWAS for privacy purposes, would it not make sense to estimate associations with each a priori or exploratory domain/class of questions, separately? Wouldn't this be the most effective way to address the specific selection/missingness mechanisms for each downstream hypothesis? This would also preempt any questions about your modelling decisions / pipeline for your full and reduced bi-factor FA or the relevance of factors that

only explain 51% and 26% of the overall non-response. SNPs poorly predicted an overall summary score, but perhaps heritability estimates would be higher if scores were aggregated in other domain/construct specific ways?

Response:

We very much appreciate these comments and the reviewer's inclination to move toward domain-specific analyses. The reviewer is of course correct that it's possible that additional insights could be gained from looking at fitted domains in more detail. We had previously not done this because, as demonstrated in our points above: (1) our scientific interest was in overall nonresponse, (2) the fitted sub-domains are likely more closely tied to UKB survey design and thus of less generalizable interest, and (3) nonresponse in those specific domains likely gets closer to inference about specific nonresponses where ethical concerns potentially outweigh scientific benefit, especially noting that many of our observed subdomains are for items with obvious sensitivities. After discussion with our co-authors and the editors, we have chosen to stick with our original analytic approach. We elaborate here on our motivation for that choice and on the merits of our analytic approach given that decision.

First the scientific motivation: Prior literature (including the work efficiently summarized by the reviewer) has carefully considered the impacts and correlates of nonresponse and selection bias, but that work has largely focused on impact on analyses of single items or within a single survey (though others have also begun to expand upon this). Identifying patterns of nonresponse that are shared across the diverse range of surveys in UKB is an opportunity to get new insights on what features of nonresponse are broadly generalizable. This is important in UKB, where analyzing pairs of traits from different domains should benefit from understanding the shared confounding those domains, and potentially provides information about which effects observed in the prior literature generalize beyond their originally studied survey domain. For those reasons, assessing a shared factor is of scientific interest despite not capturing all possible explanations for nonresponse for any given item.

Given this focus, the model of interest directly corresponds to the model of interest, with a shared factor of overall nonresponse behavior that is the quantity of interest while allowing for expected domain-specific correlation structure.

Compared to modelling each domain separately, this directly models the shared component while including as a special case that the nonresponse in the subdomains is fully distinct (i.e. loadings to the general factor are null) or the general factor is an emergent property of correlations between the domains (<https://www.tandfonline.com/doi/pdf/10.1080/00273171.2012.715555>). We are also

reassured that the bifactor model is well understood (<https://www.tandfonline.com/doi/pdf/10.1080/00273171.2012.715555>) with a solid track record of successfully handling subscales in modelling general factors with subscales in areas like psychopathology (Capsi et al. 2014, <https://onlinelibrary.wiley.com/doi/abs/10.1002/da.20432>), cognition (Kamphaus 2005) and health outcomes (https://www.researchgate.net/profile/Ronald-Hays/publication/6351610_The_Role_of_the_Bifactor_Model_in_Resolving_Dimensionality_Issues_in_Health_Outcomes_Measures/links/5863a54208aebf17d39739f7/The-Role-of-the-Bifactor-Model-in-Resolving-Dimensionality-Issues-in-Health-Outcomes-Measures.pdf). We're further reassured that we empirically show very good model fit of the bifactor model to the nonresponse data. So conditional on the decision to focus on a single overall factor, we're confident in our choice of modelling pipeline.

That leaves the question of what we might do with the domain specific factors. Empirically, our results clearly suggest meaningful domain-specific signal. The reviewer rightly infers that this likely means higher heritability could be observed for nonresponse within a domain if performed GWAS within a domain including both shared and domain-specific components (though this is not guaranteed, if e.g. the domain specific components are less heritable than the general factor). However we might anticipate that domain-specific estimation is liable to yield factors more idiosyncratic to nonresponse in UKB (e.g. for nonresponse shared across TV time, tea intake, and cell phone use. Such factors are likely to be of narrower scientific interest (though of course still useful within UKB studies).

Perhaps more importantly, domain-specific studies considering the impact of nonresponse will likely benefit from study-specific modelling and domain knowledge rather than our attempts to summarize the few domains that emerge in our survey of 100+ items. Instead we expect that our best contribution is the provide information on overall nonresponse than can then be used to improve modelling of nonresponse within any given study.

Finally, there's the ethical concerns that keep us away from investigating the effects on nonresponse within our observed subdomains. We very much appreciate the reviewer's additional insights about weighing the risks and benefits of the ethics involved, but we do believe that there is an important imperative to avoid analyses that violate participants desire for non-response. Given that our scientific interests for this project are already focused elsewhere, and that at face value the subdomains observed in our analysis represent sensitive phenotypic domains, we have respectfully chosen to prioritize these conservative ethical bounds. We are however again very grateful for

these comments, and we considered them thoroughly before deciding to stick to our original plan.

Citations

Caspi, Avshalom, et al. "The p factor: one general psychopathology factor in the structure of psychiatric disorders?." *Clinical psychological science* 2.2 (2014): 119-137.

Kamphaus, Randy W. "The Wechsler Intelligence Scale for Children—Third Edition (WISC-III)." *Clinical Assessment of Child and Adolescent Intelligence*. Springer, New York, NY, 2005. 182-230.

Comment:

A related point: The authors state that they intend to avoid analyses that violate participants' desire for non-response, which seems an admirable and ethical goal. However, how does this square with the potential use of genetic scoring to model missingness mechanisms, which in effect, attempt to restore information about the participants' response (or at least response type)? The authors allude to this challenge when they say that one issue of non-response is a reduction of the effective sample size. Presumably the restoration of effective sample size comes from imputation of plausible values for individuals with similar genetic scores on these items, or somewhat equivalently, upweighting of individuals with non-missing response but otherwise similar characteristics (including genotype)? It seems with respect to ethical considerations, in balance, it would be a greater imperative to correct the biases arising from misattributing genetic correlations, wherein say intergenerational disadvantage leads to differential participation in the genetic component and item response and thus reintroduction of population stratification due to selection/collider bias. These biases may in turn re-inforce/re-ify perceptions around the relative contribution of genetics to education, mental capacity, "compliance" or reticence to participate in research, and other inferences that may arise from less-than-careful interpretations of these genetic correlations.

Response:

We appreciate this insightful question. The reviewer is correct that many (perhaps most) possible methods for addressing selection bias will implicitly involve some form of modelling missingness for individual items in way that implies estimated values for the

response a participant has chosen to omit. The Heckman correction analysis that we've added to the paper in fact does this to some degree, though we note it doesn't require directly estimating the phenotype for nonrespondants and instead provides information without directly affecting the sample size.

The key distinction in our view is that these anticipated forms of bias correction don't expose those estimates as the focus for inference. Methods that internally estimate likelihoods or expectations for unobserved values on the way to computing the improved estimands of scientific interest have a much different risk profile than directly reporting effect sizes that link the observation of nonresponse to estimated phenotypes (e.g. we omit effect sizes from the Heckman correction selection model for precisely this reason). Providing these GWAS results for even the general nonresponse factors is not risk-free, but we share your assessment that there is a balancing point between the most cautious study ethics vs. preventing better science that would be otherwise corrupted by these biases.

We've updated Box 1 to try to more clearly address our view of the distinctions here.

Comment:

2. Further, with respect to your study objectives, why is it important that the factor score for non-response increases prediction of future participation over other "risk factors" (0.3%, at that)? Is it not sufficient that genotype associate with nonresponse including education and health as genetic "causes" of non-response may be mediated through these? I suspect that further elaboration and specification of a proposed solution to non-response would then inform the utility of these residual correlations.

Response:

The reviewer is correct that the genetics of nonresponse (and downstream concerns about GWAS bias) would be noteworthy even if the observed prediction of nonresponse was fully mediated by other risk factors. As we've now clarified in the main text, the residual prediction beyond basic risk factors matters because a) it reaffirms the genetic finding the GWAS of PNA/IDK isn't solely an underpowered GWAS of e.g., educational attainment, and b) the unexplained variance in nonresponse predicted by the risk factors suggests that correction for the risk factors would be insufficient to capture all of the relevant signal and thus could be improved by including our more direct information about nonresponse. As the reviewer correctly predicted, we now more clearly demonstrate this possible application of the nonresponse factors in our

added evaluation of Heckman correction.

Comment:

REFERENCES

Munafò MR, Tilling K, Taylor AE, Evans DM, Davey Smith G. Collider scope: when selection bias can substantially influence observed associations. *Int J Epidemiol*. 2018;47:226–235.

Griffith GJ, Morris TT, Tudball MJ, et al. Collider bias undermines our understanding of COVID-19 disease risk and severity. *Nat Commun*. 2020;11:5749.

Haworth S, Mitchell R, Corbin L, et al. Apparent latent structure within the UK Biobank sample has implications for epidemiological analysis. *Nat Commun*. 2019;10:333.

Huang JY. Representativeness Is Not Representative: Addressing Major Inferential Threats in the UK Biobank and Other Big Data Repositories. *Epidemiology*. 2021 Mar 1;32(2):189-193.

Keyes KM, Westreich D. UK Biobank, big data, and the consequences of non-representativeness. *Lancet*. 2019;393:1297.

Richiardi L, Pearce N, Pagano E, Di Cuonzo D, Zugna D, Pizzi C. Baseline selection on a collider: a ubiquitous mechanism occurring in both representative and selected cohort studies. *J Epidemiol Community Health*. 2019;73:475–480.

Smith LH, VanderWeele TJ. Bounding bias due to selection. *Epidemiology*. 2019;30:509–516.

Sohail M, Maier RM, Ganna A, et al. Polygenic adaptation on height is overestimated due to uncorrected stratification in genome-wide association studies. *Elife*. 2019;8:e39702.

Stamatakis E, Owen KB, Shepherd L, Drayton B, Hamer M, Bauman AE. Is cohort representativeness passé? Matching the UK Biobank sample to source population characteristics and recalculating the associations between lifestyle risk factors and mortality. *Epidemiology*. 2021 Mar 1;32(2):179-188.

Swanson JM. The UK Biobank and selection bias. *Lancet*. 2012;380:110.

Taylor AE, Jones HJ, Sallis H, et al. Exploring the association of genetic factors with participation in the Avon Longitudinal Study of Parents and Children. *Int J Epidemiol.* 2018;47:1207–1216.

Response:

We again thank the reviewers for this wonderful list of helpful literature, and we have now incorporated this literature into our main text.

Comment:

Data & methodology:

The study is generally well described, particularly the construction of the latent non-response factors through EFA / CFA. As alluded to above, engagement with the past literature on selection bias in UK Biobank and implications on genotypic associations is needed. These associations need not be causal: Selection bias into contribution of genetic material can also explain these findings, even if restricted to a single self-reported ancestry (European).

Response:

We have now engaged with literature thoroughly, as well as implications on genetic associations. The host of possible influences on missing data, including for example the influence of ancestry on selection bias, is now reflected in the new Figure 1 and its corresponding text.

Comment:

Additionally, more details on the LD Score Regressions and PRS constructions would be appreciated both in the main methods and as supplements. E.g. which / how many SNPs were chosen, what were the thresholds for selection, etc.

Response:

In response to the reviewer and Reviewer 1, we have made our use of LD Score Regression more clear, highlighting these results in the main text in the “Genome-wide association study (GWAS) of item nonresponse” section and in Supplementary Table 7. We have also discussed our use of LD Score Regression more specifically in the Methods in the “Heritability and enrichment” section.

For the polygenic scores in Add Health, we have updated the Methods section about score construction (“Polygenic risk scoring”), which includes details such as how many SNPs were chosen and the thresholds for selection. We include this text below for reference. Hopefully the section is now more clear, but if there are any additional details the reviewer wishes to see we’d be happy to add them.

“For the Add Health sample, we used the genotyped data from the Add Health prediction cohort to create the LD reference file. After imputing the genetic data to the Haplotype Reference Consortium (HRC) using the Michigan Imputation Server, we used only HapMap3 variants with a call rate > 98% and a minor allele frequency > 1% to construct the polygenic scores. We limited the analyses to European-ancestry individuals. Polygenic scores were calculated with an expected fraction of causal genetic markers set at 100%. In total, we used 1,168,025 HapMap3 variants to construct the polygenic scores in Add Health. We then used Plink to multiply the genotype probability of each variant by the corresponding LDpred posterior mean over all variants. In total, we created two polygenic scores, using the summary statistics of our two main phenotypes: 1) Prefer not to Answer (PNA) and 2) I Don’t Know (IDK). We then determined the association of the polygenic score for the related Refused to Answer and I Don’t Know Phenotypes in Add Health. Prediction accuracy was based on a logistic regression of the outcome phenotype on the polygenic score and a set of standard controls, which include birth year, sex, an interaction between birth year and sex, and the first 10 genetic principal components of the variance-covariance matrix of the genetic data. Variance explained by the polygenic scores was calculated in regression analyses as the Nagelkerke’s pseudo- R^2 change, i.e. the pseudo- R^2 of the model including polygenic scores and covariates minus the pseudo- R^2 of the model including only covariates. 95% confidence intervals around all pseudo- R^2 values are bootstrapped with 1000 repetitions each. We also used a recently developed score for educational attainment to predict both of our binary nonresponse outcomes in Add Health.”

Comment:

Conclusions: Clearer elaboration of the use of these genetic correlations to understanding and correcting non-response bias is needed. This will entail engaging with the selection bias literature a bit more closely. Relatedly, the authors have the potential to improve understanding of non-response bias over existing literature by making more use of domains/patterns in non-response, rather than opting for a single underlying construct.

Response:

As stated in our responses above, we have thoroughly engaged the literature, added a careful consideration of the diagramming of nonresponse bias (Figure 1), and added analyses demonstrating a Heckman-based correction of item nonresponse at the genetic and phenotypic levels. While we have refrained from heading in the direction of making use of domains/patterns in nonresponse, we have outlined our reasons for sticking with our original plan, in correspondence with our collaborators and the editors.

Reviewer #3

Comment:

Thanks for the opportunity to read the manuscript “Patterns of item nonresponse behavior to survey questionnaires are systematic and have a genetic basis”. As the title suggests, this is a study of genetically linked nonresponse patterns in order to reduce issues with MNAR in respect to latent variables.

I believe this is a very interesting contribution, well designed, conducted, and written, and it merits potentially a high-impact publication, but I would challenge the authors for two main points and two minor ones.

Response:

We thank the reviewer for the kind words about our manuscript, and we have tried our best to address the concerns below.

Comment 1:

First, item non-response is a very, very broad and well-established field of research and I believe that this study could build more on this literature on all levels. First, the main mediators of genetic effects on non-response appear to be education and health and there should be a theory about why this is the case. Second, there must be further literature (I’m not an expert and would recommend one as a reviewer) on the main predictors of item non-response which needs review, and the findings need to be related to existing knowledge. Finally, the contributions of this study which I see so far are first that it can measure a “latent” “disposition” for non-response in contrast to observed predictors. This contribution could be further quantified beyond the revisualization of education and health. It would also be interesting to see the summary statistics revisualized for the genetic correlations with other traits which are potentially measurable and the resulting heritability of the non-response GWAS. While the

theoretical contribution of receiving a yet unmeasured predictor for item-nonresponse is clear and great, it should be made very clear as well what the practical/empirical contribution is, how much variance we explain on top of measurable variables or genetically explored ones.

Response:

Based on this reviewer's comments as well as those of the editors and additional reviewers, we have now included a much more thorough review of the existing literature on nonresponse and differential participation. We appreciate Nature Human Behavior's flexibility on number of reference, etc to enable including this additional commentary and review of existing theories on e.g. why education and health reliably affect item nonresponse.

The analysis of residual genetic components has been updated based on the suggestions of all of the reviewers. First, we have now expanded the heritability residualization model to include the genetic effects of the SES variables educational attainment, income, and region-based social deprivation. The additional contribution of region-based social deprivation to NR heritability was nonsignificant beyond that of EA and income despite being anticipated to be an important correlate of nonresponse. We report the estimated heritability conditional on this model in the main text (1.35% for PNA, 5.27% for IDK). Given that power for modelling additional variables beyond this initial set is limited by this moderate residual heritability, we did not pursue further residualization of heritability but instead focus on providing more thorough interpretation of the remaining genetic correlations to other traits. Specifically, we have reformatted Supplementary Table 9, which now lists for PNA and IDK factors and 654 other traits: 1) unadjusted correlations, 2) correlations adjusted for the other NR factor, and 3) correlations adjusted for SES variables (listed above). In the Results section of the main text, we now give more examples and interpretation of these results.

Finally, if we understand the reviewer correctly there's an interest here in considering full genome-wide summary statistics for IDK and PNA conditional on SES/etc – an interest we share. Unfortunately we're concerned that such results might be affected by collider bias if the genetic overlap reflect sharing of only some biological pathways while others are unique to educational attainment/income/etc. In that case some summary statistics would be misleading, reflecting loci with difference effects on IDK/PNA and EA even when the true effect on IDK/PNA is zero (<https://www.ncbi.nlm.nih.gov/pmc/articles/PMC4320269/>). We hope that ongoing methods work will resolve that concern to allow evaluation of conditional GWAS without collider bias (e.g. <https://www.nature.com/articles/s41467-022-28119-9>) so that this

question (and many other research questions like it) can be addressed in the near future.

Comment 2:

I am really wondering about the practical value of the knowledge of this selection variable. My intuition is that it can be used for some sort of conditional/selection regression analysis a la Heckman. Can't this be demonstrated, for example, in the Add Health prediction sample, e.g. including the selection score in a Heckman model of the education score predicting educational attainment? What else will be the use?

Response:

We appreciate the reviewer's interest in not only characterizing nonresponse in GWAS but also evaluating ways to apply this information. Based on this suggestion, we have added new analyses demonstrating the practical utility of our results using a proof-of-concept Heckman correction of genetic and phenotypic association, which we highlight in the main text ("Heckman correction of GWAS results"), in Figure 6, in methods ("Heckman methods"), in the supplement ("Heckman correction of phenotypic associations", "Heckman correction of GWAS with only nonresponse factors", "Interpretation of Heckman correction results"), and finally Supplementary Tables 12, 13, 14. Although future work beyond the scope of this paper will be necessary to fully evaluate best practices for the use of this information in Heckman or other bias correction methods, we believe that this has substantively improved the potential interest in and applicability of our investigation into item nonresponse.

Comment:

Minor

3. Another very interesting feature is the high genetic correlation between IDK and PNA, as if this is partly the same trait? I was really wondering what that means – could you interpret it? Furthermore, there is a conditional genetic correlation analysis of both items with other traits which is interesting, but would it also be possible to receive some sort of ranking of these categories in a genetic correlation analysis with, for example, education? As in: IDK scores higher for education than PNA.

Response:

Indeed IDK and PNA appear to be overlapping traits. We've updated the Intro and Discussion to add further interpretation of how IDK and PNA likely reflect parts of a

spectrum of nonresponse behaviors. We also explicitly highlight the number of shared loci (2) between GWAS of the traits in addition to their genetic correlations, and describe one of those shared loci in more detail. Regarding genetic correlations, we now explicitly test using genomicSEM whether the correlations between IDK and PNA to an outside trait differ, and find that this difference is significant for 38 traits. We highlight some specific examples in the main text, and full results are provided in full in Supplementary Table 8 ("P-value Diff" column) and plotted in Figure 4c.

Comment:

4. Maybe it would be good to explicitly mention that the binary predictions are based on linear probability models.

Response:

We apologize for the confusion here. We had originally written this section intending to also look at factor scores as outcomes. As we only focus on binary outcomes for PNA and IDK in the polygenic scores section, we have appropriately updated the text to reflect logistic regression and pseudo- R^2 methods in both the appropriate main text and Methods sections.

Comment:

5. The manuscript needs careful editing.

Response:

We have all now more thoroughly gone through the manuscript with regard to editing, and we hope our changes meet the reviewer's expectations.

Decision Letter, first revision:

28th July 2022

Dear Dr Wedow,

Thank you once again for your manuscript, entitled "Patterns of item nonresponse behavior to survey questionnaires are systematic and have a genetic basis," and for your patience during the peer review

process.

Your manuscript has now been evaluated by 3 reviewers, whose comments are included at the end of this letter. Reviewers 1 and 2 also commented on the previous version of the manuscript. In light of ethical concerns raised in the previous round of review, we invited an additional reviewer with ethics expertise, Reviewer 4, to contribute an ethical review of the manuscript.

Although the reviewers find your work to be of interest, they also raise some important concerns. We are interested in the possibility of publishing your study in Nature Human Behaviour, but would like to consider your response to these concerns in the form of a revised manuscript before we make a decision on publication.

To guide the scope of the revisions, the editors discuss the referee reports in detail within the team, including with the chief editor, with a view to (1) identifying key priorities that should be addressed in revision and (2) overruling referee requests that are deemed beyond the scope of the current study. In the case of your work, it is crucial that all ethical concerns raised by Reviewer 4 are addressed in full, in the form of both a point-by-point response to the reviewer and a revised manuscript. We also ask that you include an FAQ in your Supplementary Information to address ethical concerns that the readers of your work may have.

In sum, we invite you to revise your manuscript taking into account all reviewer and editor comments. We are committed to providing a fair and constructive peer-review process. Do not hesitate to contact us if there are specific requests from the reviewers that you believe are technically impossible or unlikely to yield a meaningful outcome.

We hope to receive your revised manuscript within two months. I would be grateful if you could contact us as soon as possible if you foresee difficulties with meeting this target resubmission date.

- Include a "Response to the editors and reviewers" document detailing, point-by-point, how you addressed each editor and referee comment. If no action was taken to address a point, you must provide a compelling argument. When formatting this document, please respond to each reviewer comment individually, including the full text of the reviewer comment verbatim followed by your response to the individual point. This response will be used by the editors to evaluate your revision and sent back to the reviewers along with the revised manuscript.
- Highlight all changes made to your manuscript or provide us with a version that tracks changes.

[REDACTED]

We look forward to seeing the revised manuscript and thank you for the opportunity to review your work. Please do not hesitate to contact me if you have any questions or would like to discuss these revisions further.

Sincerely,

Charlotte Payne

Charlotte Payne, PhD
Senior Editor
Nature Human Behaviour

REVIEWER COMMENTS:

Reviewer #1:
Remarks to the Author:

The authors have been very responsive to my previous concerns, and cover them extensively in the revision. I only have a few minor remarks.

What is the outlier (top-left) in Supplementary Figure 9a? Why the rg changed from -0.4 to -1 after adjustment for SES?

As in comment #1 of reviewer #3, the authors expressed the concerns of collider bias when conducting conditional analysis on the summary statistics of IDK/PNA on SES. However, Slope-Hunter (as cited by the authors) and GSMR-mtCOJO (cited and compared by Slope-Hunter paper) both claimed they adjusted for collider bias. Could the authors clarify what are the concerns of not using them?

Reviewer #2:
Remarks to the Author:

In my previous review, I noted that there was a potential for the authors' approach to characterising genetic correlations with item non-response to more granularly understand different forms of selection bias that may be at play (relative to the etiologic mechanism under study), but that clarity was lacking in exactly what a researcher should make of their findings or how they could be practically applied (given the well-documented concerns with biased genetic associations in the UKBB).

Thanks to the authors for their robust additions and revisions specifically addressing these comments.

The current summary of the past literature, elaboration on how their approach differs from and improves on past work, and provision of a workable implementation in the form of a Heckman-style correction greatly strengthens the paper. Their justification for balancing privacy-preservation and reduction of inferences bias is also reasonable. I might argue that imputation at the domain-level can also preserve privacy by their same logic, and that it is actually desirable to correct for any idiosyncrasies of the UKBB (the first goal is internal or target validity). However, I recognise these to be relatively minor quibbles in the context of the broader paper.

To be most optimistic, I think the major contribution of this paper is to hopefully push UKBB researchers to be highly skeptical that selection biases can be tackled by a single genome- and phenome- wide approach without considering the specific genetic and behavioral sources of selection relevant to the etiologic system under study. The authors' various genetic correlation results make this clear.

Thank you for the opportunity to review this revision.

Reviewer #4:

Remarks to the Author:

Thank you for asking me to review this paper from an ethics perspective. I'm appreciative of the opportunity to do so.

The paper explores an interesting premise – that is, it aims to explore the potential reasons for why some survey participants may choose nonresponse or “I do not know” answers to survey questions in an attempt to redress issues of biased data. As we move increasingly to big data analyses in health research, many of which combine self-reported data with passive clinical and non-clinical measurements, exploring how to move towards more complete datasets and addressing issues of bias is a laudable pursuit.

Nevertheless, I have particular concerns about exploring potential reasons for why some survey participants may choose nonresponse answers through correlations with genetics, as this paper has chosen to do.

Before I describe these, it is worth noting that most of the paper is impenetrable in terms of the modes of statistical analysis. This makes it incredibly difficult for a non-stats person to examine and interpret the results. While, of course, statistics are complicated, given the caution that (I will argue below) is needed when considering this paper, black boxing the analysis to make strong conclusions (see below) is very problematic.

Ethical issues

Setting out to explore behavioural traits from the perspective of genetic determinism is, as I am sure you are aware, hugely problematic. This is because making claims about survey responses and genetics ignores the wide range of social, economic, personal, cultural, political etc reasons for why behaviour is as it is (e.g., responding a particular way to a survey question may be related to e.g. not knowing the answer, having a bad day, being frustrated with the questions etc). It is also hugely problematic because

it perpetuates discourses that reduce behaviour to genetics. Such discourses can lead to a range of social and ethical concerns, such as those relating to stigma, discrimination, and inequality/justice. Previous research that has led to discourses promoting the “gay gene” and/or “the violence gene” are examples of such reductions that have led to a range of ethical issues in these areas. Furthermore, I am sure this is not the authors’ intention, but their research fails to address the long history of sensitives of exploring relationship the relationship between behavioural traits and genetics, and this is extremely problematic.

We need to also ask what utility the findings could possibly have for insights to human health - even if some genetic basis could be robustly demonstrated. This is crucial because the use of UK Biobank data is specifically premised on the need to contribute to health (this is a key aspect of access), but also because of general questions related to the value of the results. For instance, the authors discuss how their findings can help address genetic missingness, but I was a little confused about how the authors addressed the fact that the relationship to genetics might be a red herring for other confounding variables. In my lay understanding, it reminds me of a study the authors referred to at the beginning of the paper that associated body fat with survey response (or something like that). It also makes me think about if there is a small genetic link, how does that relate to the fact that I can say I have a 50% chance of flipping heads when I randomly toss a coin? In short, I was also confused by how the polygenicity of the findings would be addressed and what this meant in terms of what value could ever really come out of these findings. In my opinion, there are just too many variables that influence behavioural traits to say anything socially/genetically useful leading me to question the value of this study.

I would urge extreme caution if you are considering publishing this article in your journal as I believe it not only holds very little value (and to me is an example of wasted resources (financially and environmentally)), but also raises ethical concerns about its reductionist approach to behavioural traits.

Author Rebuttal, first revision:

Response to Reviewer

Reviewer #1

Comment 1:

What is the outlier (top-left) in Supplementary Figure 9a? Why the r_g changed from -0.4 to -1 after adjustment for SES?

Response:

We thank the reviewer for catching what ended up being a coding error on our part. The outlier mentioned is “years completed full time education.” The coding error led to the

coloring of the dot being incorrect. We have updated the figure after fixing the coding, and have carefully gone through all our coding, for figures and for analyses, with this revision to guarantee that no other errors exist. Again, we thank the reviewer for catching this error!

Comment 2:

As in comment #1 of reviewer #3, the authors expressed the concerns of collider bias when conducting conditional analysis on the summary statistics of IDK/PNA on SES. However, Slope-Hunter (as cited by the authors) and GSMR-mtCOJO (cited and compared by Slope-Hunter paper) both claimed they adjusted for collider bias. Could the authors clarify what are the concerns of not using them?

Response:

Slope-Hunter is a fantastic advance in adjusting GWAS for collider bias, but unfortunately it doesn't allow us to confidently consider genome-wide results for nonresponse conditional on SES/etc. for two main reasons:

1) Slope-Hunter is only designed to work with LD-pruned SNPs rather than full genome-wide results. This requirement arises from Slope-Hunter performing regression with the SNPs as observations (without any modeling of correlated residuals), thus only allowing independent markers to be included. Thus Slope-Hunter doesn't enable providing a fully genome-wide set of results for conditional analysis adjusted for collider bias. (For what it's worth, it also requires as input a GWAS that was run conditioning phenotypically on the chosen confounder. The UK Biobank data would be sufficient to run this conditional GWAS, but that is different from our current nonresponse GWAS, and so adding this analysis would involve an entirely new GWAS rather than simply being an additional secondary analysis.)

2) Slope-Hunter still requires assumptions about the consistency of genetic effects for the confounder (e.g., SES) that we're not confident would be met for our application. Specifically, Slope-Hunter assumes that there is a linear relationship between the SNP's conditional GWAS beta and the SNP's effect on the confounder that is consistent across all SNPs that influence both the confounder and the outcome (assumption A2 in their paper). Slope-Hunter provides better flexibility than earlier

methods by limiting this assumption to only SNPs affecting both phenotypes, rather than all SNPs genome-wide, but it still assumes consistency of the collider bias in that set of SNPs. For our analysis, this would require assuming that every SNP that affects e.g., nonresponse that is partially mediated by observed educational attainment has the same relative bias. (GSMR-mtCOJO would require the even stronger assumption that educational attainment has a directly causal effect on nonresponse.) It seems unlikely to us that would hold; for example, we wouldn't expect controlling for EA in the GWAS of "I don't know" responses to have the same relative impact of collider bias for SNPs that affect memory and SNPs that affect conscientiousness. If we cannot make that assumption, then we cannot expect Slope-Hunter to provide unbiased estimates that fully correct for collider bias.

We therefore favor sticking to the primary GWAS of nonresponse, where the interpretation and limitations can be discussed more clearly, rather than trying to present an adjusted conditional result for only pruned loci from Slope-Hunter that runs the risk of having unknown amounts of uncorrected collider bias. That's not to suggest a Slope-Hunter analysis with the current data couldn't still be interesting, but we anticipate the necessary work to evaluate trends in what may be an incomplete correction, test sensitivity to tuning parameters (e.g., Slope-Hunter's lambda, LD pruning), etc. would not fit the scope of the current paper. So as we noted in our prior response to reviewer #3 we instead remain hopeful that further development in the area of collider bias corrections will allow us to return to this question of conditional genome-wide effects on nonresponse in the future.

Decision Letter, second revision:

21st November 2022

Dear Dr. Wedow,

Thank you for submitting your revised manuscript "Patterns of item nonresponse behavior to survey questionnaires are systematic and have a genetic basis" (NATHUMBEHAV-22010117B). We really appreciate your response to editorial and reviewer concerns and the revisions you have made to your manuscript. We will therefore be happy in principle to publish it in Nature Human Behaviour, pending minor revisions to comply with our editorial and formatting guidelines.

We are now performing detailed checks on your paper and will send you a checklist detailing our editorial and formatting requirements within a week. Please do not upload the final materials and make any revisions until you receive this additional information from us.

Sincerely,

Charlotte Payne

Charlotte Payne, PhD
Senior Editor
Nature Human Behaviour

Final Decision Letter:

Dear Dr Wedow,

We are pleased to inform you that your Article "Patterns of item nonresponse behavior to survey questionnaires are systematic and associated with genetic loci", has now been accepted for publication in Nature Human Behaviour.

Please note that *Nature Human Behaviour* is a Transformative Journal (TJ). Authors whose manuscript was submitted on or after January 1st, 2021, may publish their research with us through the traditional subscription access route or make their paper immediately open access through payment of an article-processing charge (APC). Authors will not be required to make a final decision about access to their article until it has been accepted. IMPORTANT NOTE: Articles submitted before January 1st, 2021, are not eligible for Open Access publication. Find out more about Transformative Journals

With best regards,

Charlotte Payne

Charlotte Payne, PhD
Senior Editor
Nature Human Behaviour